# PRETRAINED DEEP MODELS OUTPERFORM GBDTS IN LEARNING-TO-RANK UNDER LABEL SCARCITY

## ABSTRACT

While deep learning (DL) models are state-of-the-art in text and image domains, they have not yet consistently outperformed Gradient Boosted Decision Trees (GBDTs) on tabular Learning-To-Rank (LTR) problems (Qin et al., 2021). Most of the recent performance gains attained by DL models in text and image tasks have used unsupervised pretraining (Devlin et al., 2018; Chen et al., 2020), which exploits orders of magnitude more unlabeled data than labeled data. To the best of our knowledge, unsupervised pretraining has not been applied to the LTR problem, which often produces vast amounts of unlabeled data.

In this work, we study whether unsupervised pretraining of deep models can improve LTR performance over GBDTs and other non-pretrained models. By incorporating simple design choices–including SimCLR-Rank, an LTR-specific pretraining loss–we produce pretrained deep learning models that consistently (across datasets) outperform GBDTs (and other non-pretrained rankers) in the case where there is more unlabeled data than labeled data. This performance improvement occurs not only on average but also on outlier queries. We base our empirical conclusions off of experiments on (1) public benchmark tabular LTR datasets, and (2) a large industry-scale proprietary ranking dataset. Code is provided at https://anonymous.4open.science/r/ltr-pretrain-0DAD/README.md.

## 1 INTRODUCTION

The learning-to-rank (LTR) problem aims to train a model to rank a set of items according to their relevance or user preference (Liu, 2009). An LTR model is typically trained on a dataset of queries and associated *query groups* (i.e., a set of potentially relevant *documents* or *items* per query), as well as an associated (generally incomplete) ground truth ranking of the items in the query group. The model is trained to output a ranking of documents or items in a query group, given a query. LTR is a core problem in many real world applications—most notably in search contexts including Bing web search (Qin & Liu, 2013), Amazon product search (Yang et al., 2022), and Netflix movie recommendations (Lamkhede & Kofler, 2021).

In many applications of LTR, models take as input *tabular features*—numerical or categorical features—of queries and documents (Chapelle & Chang, 2011; Qin & Liu, 2013; Lucchese et al., 2016). Today, deep models are largely outperformed by gradient boosted decision trees (GBDTs) (Friedman, 2001) over tabular features (Jeffares et al., 2023; Qin et al., 2021). In contrast, deep models are state-of-the-art by a significant margin in domains like text (Devlin et al., 2018) and images (He et al., 2016).

Recent breakthroughs in modeling non-tabular data like text and images have been driven by first training a deep neural network to learn from unlabeled data (unsupervised pretraining, or pretraining) (Devlin et al., 2018; Chen et al., 2020), followed by supervised training (finetuning). Models that are pretrained in this way can perform significantly better than models that were only trained on existing labeled data, which is often limited in size. The remarkable success of unsupervised pretraining in the image and text domains over plain supervised deep learning appears to arise in part from two factors: (1) there exist large, available sources of unlabeled text and image data, and (2) pretrained models are able to take advantage of unlabeled data.

A natural question is whether deep models can outperform tabular methods like GBDTs on the LTR problem by making use of unsupervised pretraining. Note that GBDTs (to the best of our knowledge)

are unable to make use of unsupervised pretraining. In this work, we show that the answer to this question is yes, as long as labels are scarce or sparse (scarce means there are few labeled query groups while sparse means each query groups has few labels). We base our empirical conclusions off of experiments on three well-known public datasets for ranking: MSLRWEB30K (Qin & Liu, 2013), Yahoo (Chapelle & Chang, 2011), Istella (Lucchese et al., 2016), and a large industry-scale proprietary ranking dataset.

**Contributions:** *(1) We demonstrate that unsupervised pretraining can produce deep models that outperform GBDTs in ranking.* Pretrained deep rankers outperform GBDTs signficantly with respect to NDCG (normalized discounted cumulative gain) (Burges, 2010) when labels are scarce and/or sparse. *(2) We provide empirically justified LTR-specific pretraining strategies, including a new ranking-specific pretraining loss, SimCLR-Rank.* We find that unsupervised pretraining behaves differently in LTR vs other settings (like images or text), and so requires its own set of strategies. *(3) We demonstrate that pretrained deep rankers can perform better than GBDTs on outlier query groups when labels are scarce and/or sparse.*

**Related work:** We focus on the traditional LTR setting where the features are all numeric (tabular data). In this setting, gradient boosted decision trees (GBDTs) (Friedman, 2001) are the de-facto models, and deep models have yet to outperform them convincingly (Qin et al., 2021; Joachims, 2006; Ai et al., 2019; Bruch et al., 2019; Ai et al., 2018; Pang et al., 2020; McElfresh et al., 2023). Self-supervised learning (SSL) or unsupervised pretraining has improved overall performance and robustness to noise (Hendrycks et al., 2019) in settings where there is a significant source of unlabeled data like text (Devlin et al., 2018) and images (Chen et al., 2020).

Please refer to Appendix A.1 for a detailed exposition of related work.

## 2 LEARNING-TO-RANK AND ITS METRICS

The training data in LTR consists of $n$ query groups (QGs). The $i$-th *query group* consists of $L_i$ potentially relevant items for the query (e.g. products) represented by feature vectors $\boldsymbol{x}_{i,j} \in \mathbb{R}^d$, and relevance labels $y_{i,j}$ which could be binary, ordinal, or real-valued measurements of relevance (Qin et al., 2021). Altogether, the training data is written as $D = \{\{x_{i,j}\}_{j=1}^{L_i}, \{y_{i,j}\}_{j=1}^{L_i}\}_{i=1}^n$. The objective is to learn a function that, given a query group $k$, ranks the $L_k$ items $\{x_{k,j}\}_{j=1}^{L_k}$ such that the items with highest relevance are ranked at the top. In this paper, we consider unsupervised pretraining, so we also have a larger unlabeled dataset $D'$, which contains $m$ query groups where $D' = \{\{x_{i,j}\}_{j=1}^{L_i}\}_{i=1}^m$ where $m \geq n$ and the query groups of $D$ are a subset of those in $D'$. Most LTR algorithms formulate the problem as learning a scoring function $f_\theta : \mathbb{R}^d \to \mathbb{R}$ that maps the feature vector associated with each item to a score, and then ranks the items by sorting the scores in descending order.

To measure the quality of a ranking induced by our scoring function $f_\theta$ on the $k$-th query group, a commonly-used metric is NDCG: $\text{NDCG}(\pi_s, \{y_{k,j}\}_{j=1}^{L_k}) = \text{DCG}(\pi_s, \{y_{k,j}\}_{j=1}^{L_k})/\text{DCG}(\pi^*, \{y_{k,j}\}_{j=1}^{L_k})$, where $\pi_s : [L_k] \to [L_k]$ (where $[L]$ is the list $\{1, \ldots, L\}$) is a ranking of the $L_k$ elements of the $k$th query group induced by the scoring function $f_\theta$ on $\{x_{k,j}\}_{j=1}^{L_k}$ while $\pi^*$ is the ideal ranking induced by the relevance labels $\{y_{k,j}\}_{j=1}^{L_k}$, and discounted cumulative gain (DCG) is defined as $\text{DCG}(\pi, \{y_{k,j}\}_{j=1}^{L_k}) = \sum_{j=1}^n \frac{2^{y_{k,j}} - 1}{\log_2(1 + \pi(j))}$. Typically, a truncated version of NDCG is used that only considers the top-$u$ ranked items, denoted as NDCG@$u$. In the rest of our paper, we will refer to NDCG@5 as NDCG, and this will be the main evaluation metric we consider.

### 2.1 OUTLIER-NDCG FOR OUTLIER PERFORMANCE EVALUATION

In interactive ML systems like search, performing well on outlier queries is particularly valuable as it empowers users to search for more outlier queries, which in turn allows the practitioner to collect more data on outliers and improve the model. To this end, we design a metric that evaluates NDCG only on outlier queries dataset. In practice, outlier queries may already be known, and the

practitioner can define the outlier datasets accordingly. For example, since industry data pipelines often have missing data/features one could identify samples with missing features as outliers.

When the outliers are not already known, defining outliers (particularly for high-dimensional data) is challenging. Hence, we build an outlier dataset using the following intuition: outliers are rare values that are separated from most of the data. We systematically select outlier query groups, with details given in Appendix A.2.1. Outlier-NDCG is defined as the NDCG on the outlier query groups.

## 3 DESIGN: UNSUPERVISED PRETRAINING FOR LTR

This section presents the DL pretraining baselines we consider in this work, **including our proposed LTR-specific pretraining loss, SimCLR-Rank**. We ultimately find that the best pretraining strategy can depend on the dataset, so we avoid prescribing any of these baselines across the board. We start by first presenting SimSiam and SimCLR, two of the best-known pretraining approaches from the class of *contrastive learning* methods (Chen et al., 2020; Chen & He, 2021) in LTR context.

*SimCLR* (Chen et al., 2020). For a data point $x_{i,j}$ ($i$-th query group, $j$-th item in the query group) in a batch, we produce stochastically augmented versions $x_{i,j}^{(1)}$ and $x_{i,j}^{(2)}$ which are called a *positive pair*. Second, a base encoder $h(\cdot)$ and projection head $g(\cdot)$ map $x_{i,j}^{(1)}$ to $z_{i,j}^{(1)} = g(h(x_{i,j}^{(1)}))$ and $x_{i,j}^{(2)}$ to $z_{i,j}^{(2)} = g(h(x_{i,j}^{(2)}))$. Then we optimize the InfoNCE loss (Oord et al., 2018) to push $z_{i,j}^{(1)}$ and $z_{i,j}^{(2)}$ closer to each other and $z_{i,j}^{(1)}$ farther from other augmented data points in the batch, both in cosine similarity (Chen et al., 2020). Let $B$ be the number of query groups in the batch, $\tau$ a temperature parameter, and $\text{sim}(a,b) = \langle a,b \rangle / \|a\|\|b\|$ represent the cosine similarity. Precisely, the SimCLR loss is as follows with respect to the $x_{i,j}^{(1)}$ (with a corresponding loss for $x_{i,j}^{(2)}$):

$$\ell_{i,j}^{(1)} = -\log \frac{\exp(\text{sim}(z_{i,j}^{(1)}, z_{i,j}^{(2)})/\tau)}{\sum_{q=1}^{B} \sum_{k=1}^{L_q} \sum_{u=1}^{2} \mathbb{1}\{(q,k,u) \neq (i,j,1)\}[\exp(\text{sim}(z_{i,j}^{(1)}, z_{q,k}^{(u)})/\tau)]} \tag{1}$$

SimCLR enjoys wide adoption, because of its superior performance in many domains (like images) (Chen & He, 2021; Wang et al., 2022b;a). This is because contrasting an item $A$ with another item $B$ where $B$ is difficult to distinguish from $A$ (i.e., $B$ is a "hard negative") can help a model learn good representations (Robinson et al., 2020; Oh Song et al., 2016; Schroff et al., 2015; Harwood et al., 2017; Wu et al., 2017; Ge, 2018; Suh et al., 2019). SimCLR simply contrasts against all other data in a batch including from other query groups. A large enough batch (Chen et al., 2020) is likely to contain a hard negative.

*SimSiam* (Chen & He, 2021) similarly takes a data point $x_{i,j}$ and produces stochastically-augmented versions $x_{i,j}^{(1)}$ and $x_{i,j}^{(2)}$, which are called a *positive pair*. We pass the first sample of the pair through the base encoder $h(\cdot)$, projector $g(\cdot)$, and predictor $\text{pred}(\cdot)$, to get $p_{i,j}^{(1)} = \text{pred}(g(h(x_{i,j}^{(1)})))$; we pass the second sample of the pair through just the base encoder and projector to get $z_{i,j}^{(2)} = g(h(x_{i,j}^{(2)}))$. Then we maximize $\text{sim}(p_{i,j}^{(1)}, z_{i,j}^{(2)})$. Unlike SimCLR, there are no "negative" pairs, i.e., the loss function does not try to push the representation of $z_{i,j}^{(1)}$ farther from other samples' augmentations.

SimSiam is much faster and GPU-space-efficient than SimCLR because it does not perform any negative comparisons. The time/space complexity for SimSiam is only $O(BL)$ per batch, while the time/GPU-space complexity of SimCLR is $O(B^2 L^2)$. Practically in our experiments, we find that SimCLR can be more than 100x slower than SimSiam (Table 6).

### 3.1 SIMCLR-RANK: A PRETRAINING LOSS FOR LTR

Motivated by (1) SimCLR's high complexity and (2) the efficacy of contrasting with hard negatives, we propose a third baseline, SimCLR-Rank: a pretraining loss for LTR, in place of SimCLR. Recall that the SimCLR loss (Equation (1)) samples a batch for hard negatives. However, in LTR, the hard negatives are given: *they are items from the same query group.* Therefore, we propose SimCLR-Rank, which modifies SimCLR to contrast only with items in the same query group. We give

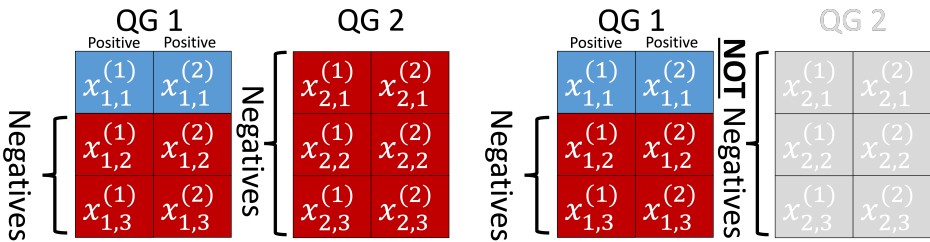

(a) In SimCLR, each positive pair is contrasted with **all other items in the batch**.

(b) In SimCLR-Rank, each positive pair is contrasted with **only items in the same QG**.

Figure 1: The difference between SimCLR and SimCLR-Rank in an example where the training batch contains two query groups (QGs) each with three items. The loss is formally given in Equation (2).

the formal loss with respect to the $x_{i,j}^{(1)}$ (with a corresponding loss for $x_{i,j}^{(2)}$) in Equation (2) and an illustrative example in Figure 1. The SimCLR-Rank loss has a time and space complexity of $O(BL^2)$ (compared to SimCLR's $O(B^2L^2)$) and SimSiam's $O(BL)$), and is empirically much faster than SimCLR (Table 6).

$$\ell_{i,j}^{(1)} = -\log \frac{\exp(\mathrm{sim}(z_{i,j}^{(1)}, z_{i,j}^{(2)})/\tau)}{\sum_{k=1}^{L_i} \sum_{u=1}^{2} \mathbb{1}[(k,u) \neq (j,1)] \exp(\mathrm{sim}(z_{i,j}^{(1)}, z_{i,k}^{(u)})/\tau)} \tag{2}$$

## 4  EMPIRICAL EVALUATION

We next evaluate these methods over public datasets and a large-scale online shopping dataset.

### 4.1  PUBLIC DATASETS

In the public datasets (MSLRWEB30K, Yahoo Set1, Istella_S), we evaluate two variants of label scarcity. In the "Relevance Score Setting", we compare pretrained vs non-pretrained models when the *number* of labeled query groups may be limited in the training set, but *within* a labeled query group, all features have relevance labels. This models the setting where practitioners are able to manually label some (but not all) training query groups.

Second, in the "Binary Label Setting", we compare these models when the training set contains stochastic binary labels instead of relevance scores. In many applications, such as online shopping and search, it is easier to obtain noisy binary labels (click, purchase, view) that come from user behavior rather than manually labeled relevance scores for the full query group. We model the binary labels as noisy observations of the relevance scores, as done in Yang et al. (2022). We detail the process in Section 4.1.2. This models the setting where practitioners can only obtain binary labels for ranking signal, which are noisy observations of the true relevance scores.

### 4.1.1  RELEVANCE SCORE SETTING

**Dataset.** We compare pretrained rankers, non-pretrained DL rankers, and GBDTs on public datasets (MSLRWEB30K, Yahoo Set1, and Istella_S) in Figure 2 and Figure 7. We vary what fraction of query groups in the training set are labeled in $\{0.001, 0.002, 0.005, 0.1, 0.5, 1.0\}$ to simulate real-world datasets with some fraction of unlabeled data. We provide dataset statistics in Table 9.

**Methodology.** *Pretraining*: (1) the encoder used for pretraining is the tabular ResNet (Gorishniy et al., 2021) with three ResNet blocks, with the final linear layer removed,(2) pretraining is done on the entire dataset with learning rate 0.0005 using Adam (Kingma & Ba, 2014) with either SimSiam or SimCLR-Rank (3) we tried four different augmentations for each pretraining method: randomly zeroing out features ("zeros") with probabilities 0.1 or 0.7, and Gaussian noise with scale 1.0 or

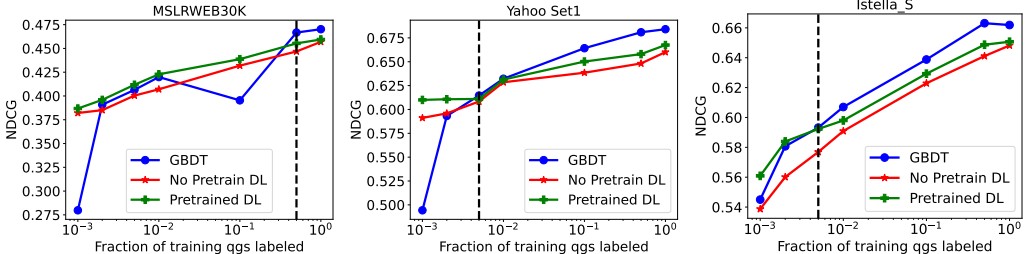

Figure 2: **Main result (Relevance score setting):** When labeled query groups have full relevance scores, for small enough fractions of labeled query groups, pre-trained DL rankers (sometimes significantly) outperform both GBDTS and non-pretrained DL rankers. We compare NDCG ($\uparrow$) as we sweep the percentage of training query groups that are labeled. To the left of the black dotted line, pretrained rankers perform the best. Points are averages over 3 trials. For all points left of the black dotted vertical line, pretrained rankers are significantly better than their non-pretrained counterparts at the $p = 0.05$ level using a one-sided $t$-test.

Table 1: Train and validation dataset statistics on MSLRWEB30K, Yahoo Set1, and Istella_S after binary label generation with $\tau_{\text{target}} = 4.5$. The test set is left untouched. Labeled QGs are those query groups that have at least one item with $y = 1$.

| Dataset | # query groups | | # labeled QGs | | # items per QG | | # positives per QG | |
|---|---|---|---|---|---|---|---|---|
| | train | val | train | val | train | val | train | val |
| MSLRWEB30K | 18151 | 6072 | 8.9% | 8.9% | 124.35 | 122.38 | 1.1% | 1.1% |
| Yahoo Set1 | 14477 | 2147 | 5.3% | 5.4% | 30.02 | 30.32 | 5.9% | 5.4% |
| Istella_S | 19200 | 7202 | 25.2% | 25.2% | 106.36 | 94.98 | 1.3% | 1.3% |

2.0, (4) we pretrain for 300 epochs, and (5) we use a batch size of roughly 200000 items (may vary based on query group size). *Finetuning*: (1) finetuning is done on the labeled train set by adding a three-layer MLP to the top of the pretrained model and training only this head for 100 epochs and then fully finetuning for 100 epochs using Adam with a learning rate of 5e-5, (2) we use an average batch size of roughly 1000 items (may vary based on query group size), (3) we use the LambdaRank loss (Burges, 2010), (4) we use the validation set to perform early stopping (i.e. using the checkpoint that performed best on the validation set to evaluate on the test set). *Hyperparameter tuning*: For pretrained rankers, in each data point we hyperparameter tune among pretrained rankers. The relevant hyperparameters are choice of pretraining method (i.e., SimCLR-Rank vs. SimSiam) and choice of data augmentation.

The DL ranker uses a 3-layer tabular ResNet from (Gorishniy et al., 2021), and was trained for 300 epochs on the labeled training set with learning rate 0.001 using Adam, with early stopping using the validation set. The GBDT ranker is the one from LightGBM (Ke et al., 2017) and we grid search the number of leaves in [31, 96, 200] and minimum data in leaf in [20, 60, 200] individually for each data point (for a total of 9 difference choices), while letting the rest of the parameters be the default in LightGBM (our tuning strategy is similar to Qin et al. (2021)). The reported values for Outlier-NDCG are those achieved by the rankers reported in Figure 2.

**Results.** We find that pretrained rankers outperform non-pretrained methods (including GBDTs) on NDCG and Outlier-NDCG across all public datasets, up to a dataset-dependent fraction of QGs labeled, as shown in Figure 2 (for NDCG) and Figure 7 (for Outlier-NDCG).

### 4.1.2 BINARY LABEL SETTING

We now evaluate pretrained rankers vs non-pretrained rankers on the binary label setting.

**Dataset.** We generate binary labels from the relevance labels of MSLRWEB30K, Yahoo Set1, and Istella_S using the methodology from Yang et al. (2022). We generate a binary label $y$ from a

Table 2: **Main Result (Binary label setting):** When $X\%$ of query groups are (binary) labeled, pretrained DL methods outperform both GBDTs and DL methods without pretraining. We generate binary labels with $\tau_{\text{target}} = 4.5$ (Section 4.1.2) on NDCG averaged over three trials. ♣ indicates when pretrained models are significantly better than non-pretrained models as measured by a t-test with significance $p < 0.05$.

| Method | MSLR (↑) | Yahoo Set1 (↑) | Istella (↑) |
|---|---|---|---|
| GBDT | $0.3335 \pm 0.0000$ | $0.6168 \pm 0.0000$ | $0.6024 \pm 0.0000$ |
| No Pretrain DL | $0.3235 \pm 0.0005$ | $0.6103 \pm 0.0072$ | $0.6084 \pm 0.0016$ |
| Pretrained DL | $\mathbf{0.3558 \pm 0.0044}$♣ | $\mathbf{0.6223 \pm 0.0008}$♣ | $\mathbf{0.6131 \pm 0.002}$♣ |
| | MSLR Outlier (↑) | Set1 Outlier (↑) | Istella Outlier (↑) |
| GBDT | $0.2885 \pm 0.0000$ | $0.5397 \pm 0.0000$ | $0.6785 \pm 0.0000$ |
| No Pretrain DL | $0.2694 \pm 0.0289$ | $0.5139 \pm 0.0053$ | $0.6874 \pm 0.0148$ |
| Pretrained DL | $\mathbf{0.2889 \pm 0.0116}$ | $\mathbf{0.5426 \pm 0.0017}$♣ | $\mathbf{0.6991 \pm 0.0027}$ |

relevance score $r$ using the following: $y = \mathbb{1}\{t \cdot r + G_1 > t \cdot \tau_{\text{target}} + G_0\}$ where $t$ is a temperature parameter, $G_1, G_0$ are standard Gumbels, and $\tau_{\text{target}}$ is a parameter controlling how sparse the binary labels are. Yang et al. (2022) show that $y$ is 1 with probability $\sigma(t \cdot (r - \tau_{\text{target}}))$ where $\sigma$ is the sigmoid function. We set $t = 4$ as in Yang et al. (2022). For a given $\tau_{\text{target}}$, we produce produce a new dataset for each of MSLRWEB30K, Yahoo Set1, and Istella_S where we convert the relevance scores in the training/validation sets to binary labels and keep the relevance scores in the test set. This models the setting where we observe binary labels that are noisy observations of true relevance scores, and we want to use these noisy observations to learn rankers that can perform well on relevance scores. We present data statistics for $\tau_{\text{target}} = 4.5$ in Table 1. After converting labels to binary with $\tau_{\text{target}} = 4.5$, we find that the realized fraction of labeled QGs in MSLR is 8.9%, in Yahoo Set1 it is 5.3%, and in Istella_S it is 25.2%.

**Methodology.** We reuse the methodology of Section 4.1.1, except we use a scoring head of one linear layer (as opposed to a three layer MLP) for pretrained models. We found that this improved stability and performance in the binary label setting.

**Results.** We present results for $\tau_{\text{target}} = 4.5$ in Table 2, showing that pretrained rankers help in binary label setting pretrained rankers also improve significantly over non-pretrained rankers. Note that by default, in LightGBM, GBDT ranker training has no randomness. We observe that even with up to 25.2% of training query groups labeled in the dataset (the results for Istella_S) pretrained rankers can perform significantly better than non-pretrained rankers. We also present results on $\tau_{\text{target}} \in \{4.25, 5.1\}$ in Appendix A.2.6. In these results, we find that making the labels sparser ($\tau_{\text{target}} = 5.1$) increases the performance improvement of pretrained rankers over non-pretrained rankers, while if labels are less sparse ($\tau_{\text{target}} = 4.25$) GBDTs are the best model.

## 4.2 LARGE-SCALE ONLINE SHOPPING DATASET

We evaluate pretraining in LTR on an internal dataset derived on a large industry-scale dataset derived from online shopping logs.

**Dataset.** The dataset is derived from online shopping logs from a large online retailer, numbering in millions of query groups. In this dataset, query groups consist of items that a shopper sees when they enter a search query. We assign two different kinds of labels to items. The first is a purchase label: we label the item as 1 if the shopper purchased an item and 0 if the shopper did not purchase the item. When using these labels to evaluate a model, we use "purchase NDCG", which is NDCG where the purchased label is used as the target gain value. Our second kind of label is a relevance label: these are hand-annotated relevance labels given to query groups that shoppers have seen. The metric we use when evaluating on relevance labels is "Relevance NDCG", where the NDCG is measured based on the relevance labels.

**Methodology.** SimSiam and SimCLR-Rank were both first pretrained unlabeled data before fine-tuned on labeled data, which was labeled using purchase labels. The production model was only trained on labeled data. Next we detail how we produced the Outlier-NDCG metrics. Internally

Table 3: Results on a large industry-scale proprietary online shopping dataset. Here we give percentage improvement of rankers pretrained by SimSiam and SimCLR-Rank . Unsupervised pre-training (1) significantly improves both overall (full dataset) performance (internally, 1% is a big improvement), and (2) *substantially* improves performance on outliers.

| Target | Pretraining method | $\Delta$% NDCG | $\Delta$% Outlier-NDCG |
|---|---|---|---|
| Purchase | Baseline | +0.00% | +0.00% |
| | SimSiam | **+1.75% $\pm$ 0.13%** | **+29.66% $\pm$ 0.84%** |
| | SimCLR-Rank | +0.18% $\pm$ 0.13% | +2.19% $\pm$ 0.71% |
| Relevance | Baseline | +0.00% | +0.00% |
| | SimSiam | **+2.78% $\pm$ 0.06%** | **+26.68% $\pm$ 0.35%** |
| | SimCLR-Rank | +0.85% $\pm$ 0.06% | +2.99% $\pm$ 0.31% |

(on our proprietary online shopping dataset), there are known feature outliers that cause low quality predictions. These outliers arise from noise introduced in various stages in the data pipeline. We have mitigation strategies in place against these feature outliers, but they are often ad-hoc and not perfect. Because we already know some of the outliers, we use them to calculate Outlier-NDCG, rather than outliers generated from the outlier detection algorithm detailed in Section 2.1. These outliers comprise 5% of the full eval dataset, which is still a large number of query groups (much larger than the number of query groups in the outlier datasets of the public datasets). We compiled an outlier test set for Outlier-NDCG using these outliers for the empirical evaluation in this paper. Our results are produced *on top* of outlier mitigation strategies–i.e. the pretrained models and the production model all use the outlier mitigation strategies.

**Results.** Our results are summarized in Table 3. The first interesting result is that SimSiam is able to get (1) significantly better results on NDCG (1% improvements on NDCG are significant internally), and (2) **significant improvements in Outlier-NDCG**—we see improvements of nearly 30% on outliers. Second, we see that both SimSiam and SimCLR-Rank improve over the internal production model. Third, here we see performance that is significantly quantitatively different between SimSiam and SimCLR-Rank. We investigate this last point in an ablation study later (Section 4.3.3).

## 4.3 JUSTIFICATION FOR DESIGN CHOICES

### 4.3.1 FULL FINETUNING IS BETTER THAN LINEAR PROBING

A popular finetuning strategy in text and images is linear probing, where one only updates a linear head on top of a pretrained model during supervised finetuning (Chen & He, 2021; Chen et al., 2020; Peters et al., 2019). It is more efficient than full finetuning (where the entire model is updated during finetuning), and can (1) help with distribution shift (Kumar et al., 2022), (2) sometimes even perform better than full finetuning! We investigate the performance of linear probing and full finetuning as finetuning strategies in LTR in this subsection.

**Dataset and Methodology.** We use the public datasets in Section 4.1.1, and let 0.2% of training query groups be labeled. We use the methodology in Section 4.1.1, and vary the finetuning strategy. When linear probing, we freeze the pretrained model and update only a linear head on top of it for 200 epochs. In multilayer probing, we freeze the pretrained model and update a 3-layer MLP head on top of it for 200 epochs. In full finetuning, we use the finetuning strategy from Section 4.1.1. Once again we use the validation set for early stopping. We compile the results in Table 4 (for SimCLR-Rank) and Table 7 (for SimSiam).

**Results.** We find that in both SimCLR-Rank and SimSiam, (1) linear probing performs poorly, and (2) MLP probing performs moderately well on SimCLR-Rank while it performs very poorly on SimSiam (aside from Yahoo Set1). By plotting the embeddings generated by the encoders (Figure 3), we offer the following explanations for the two phenomena we observe. (1) MLP probing works better for SimCLR-Rank than SimSiam because SimCLR-Rank's embeddings are more spread out, providing the opportunity for a multilayer scoring head to use the embeddings directly to distinguish items in a query group. (2) The embeddings produced by the SimSiam/SimCLR-Rank encoders are not sorted by relevance score in the projection space (unlike the fully supervised encoder, which was

Table 4: SimCLR-Rank under different finetuning strategies on NDCG/Outlier-NDCG, averaged over 3 trials. Multilayer probing performs moderately worse than full finetuning, and linear probing performs much worse than all other strategies.

| Method | MSLR ($\uparrow$) | Yahoo Set1 ($\uparrow$) | Istella ($\uparrow$) |
|---|---|---|---|
| Linear probing | $0.3219 \pm 0.0224$ | $0.5202 \pm 0.0093$ | $0.4029 \pm 0.0073$ |
| Multilayer probing | $0.3890 \pm 0.0011$ | $0.5942 \pm 0.0016$ | $0.5813 \pm 0.0006$ |
| Full finetuning | $\mathbf{0.3959 \pm 0.0022}$ | $\mathbf{0.6022 \pm 0.0013}$ | $\mathbf{0.5839 \pm 0.0013}$ |

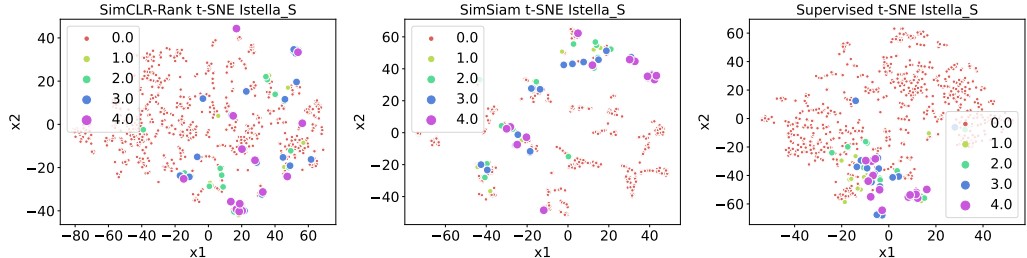

Figure 3: We plot the t-SNE plots of embeddings produced by three encoders: (1) pretrained by SimCLR-Rank, (2) pretrained by SimSiam, (3) trained via supervised training on the entire training set on roughly 1000 samples from Istella_S. Marker size and color indicates relevance. We find (1) SimCLR-Rank/SimSiam cluster different relevances effectively but do not order them as well as the supervised encoder. (2) SimCLR-Rank produces more spread-out embeddings than SimSiam.

trained on all the training set labels. This suggests that a linear ranker may not be able to directly use the embeddings to predict relevance, so full finetuning may be needed.

### 4.3.2 SimCLR-Rank outperforms SimCLR at lower cost

In this section, we show that SimCLR rank performs as well or better than SimCLR at lower cost.

**Dataset.** We evaluate on MSLRWEB30K, Yahoo's Set1, and Istella_S (Qin & Liu, 2013; Chapelle & Chang, 2011; Lucchese et al., 2016), where we remove the labels for all but 0.2% of query groups in the training set to simulate label scarcity.

**Methodology.** We follow the methodology in Section 4.1.1 except we pretrain for 20 epochs and use a batch size of roughly 2000 items (we perform a smaller scale experiment because SimCLR's computational and space efficiency prevent us from increasing the scale). During finetuning on 0.2% of the training set we finetune only the scoring head for 100 epochs and then fully finetune for 100 epochs afterwards. For hyperparameter tuning, NDCG results are reported for the best augmentation on each dataset on NDCG, and Outlier-NDCG results are reported for the best augmentation for each dataset on NDCG, not Outlier-NDCG.

**Result.** We show in Table 5 that SimCLR-Rank is slightly better than SimCLR in NDCG (it wins on 2/3 datasets), and better on 3/3 datasets on Outlier-NDCG. Given the significant efficiency gain and the slightly better performance we opt to use SimCLR-Rank over SimCLR as our representative pretraining method with negative pairs.

**Remark on Istella's Outlier-NDCG > NDCG.** In Istella, Outlier-NDCG values were higher than NDCG values for every method we tried. Istella has features for which most samples have a value of zero (Figure 5). We hypothesize that zero represents a missing data value for the feature, which would make outliers in Istella easier to rank than the "normal" values. Despite this, Outlier-NDCG still gives valuable feedback, i.e. how does the ranker perform on non-missing features? We thus continue to provide Outlier-NDCG metrics for Istella in the paper.

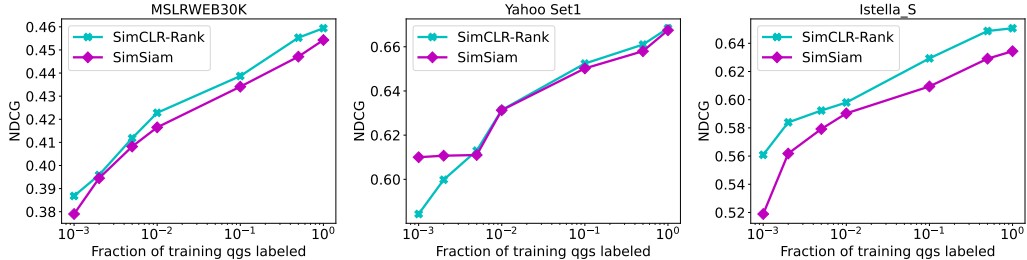

Figure 4: We plot the performance of SimCLR-Rank vs SimSiam across different fraction of training qgs labeled in the training set. Data points are averages over 3 trials. SimCLR-Rank performs better on MSLRWEB30K and Istella_S while SimSiam performs significantly better on Yahoo Set1 when data is more scarce.

Table 5: SimCLR vs SimCLR-Rank on NDCG and Outlier-NDCG, averaged over 3 trials. We keep 0.2% of labels for finetuning. SimCLR-Rank both is (1) much more efficient, and (2) produces (slightly) better rankers. ♣ is where SimCLR or SimCLR-Rank is significantly better than the other as measured by a t-test with significance $p < 0.05$.

| Method | MSLR (↑) | Yahoo Set1 (↑) | Istella (↑) |
|---|---|---|---|
| SimCLR | $0.3909 \pm 0.0021$ | $\mathbf{0.6011 \pm 0.0017}$ | $0.5829 \pm 0.0017$ |
| SimCLR-Rank (Ours) | $\mathbf{0.3929 \pm 0.0018}$ | $0.5989 \pm 0.0010$ | $\mathbf{0.5830 \pm 0.0013}$ |

| | MSLR Outlier (↑) | Set1 Outlier (↑) | Istella Outlier (↑) |
|---|---|---|---|
| SimCLR | $0.2941 \pm 0.0149$ | $0.5003 \pm 0.0050$ | $0.6051 \pm 0.0097$ |
| SimCLR-Rank (Ours) | $\mathbf{0.2964 \pm 0.0026}$ | $\mathbf{0.5128 \pm 0.0123}$ | $\mathbf{0.6305 \pm 0.0077}$ |

### 4.3.3 SimCLR-Rank vs. SimSiam

Here we perform a hyperparameter study comparing SimCLR-Rank (which we have shown to outperform vanilla SimCLR) with SimSiam.

**Dataset and Methodology.** We perform an empirical evaluation comparing SimCLR-Rank and SimSiam. The dataset and methodology follows that of Section 4.1.1, except we let (1) SimCLR-Rank use the gaussian augmentation with scale 1, and (2) SimSiam use the zeros augmentation with probability 0.1 always (these augmentations generally perform well for each method).

**Results.** In Figure 4 we find that SimCLR-Rank performs better on MSLRWEB30K and Istella_S, while SimSiam performs much better on Yahoo Set1 when the fraction of labeled training query groups is lower. By looking at the t-SNE embedding plots for Yahoo Set1(Figure 6), we see that SimSiam's more aggressive clustering strategy is largely able to cluster similar relevances together on this specific dataset, compared to SimCLR-Rank's more conservative clustering.

### 4.3.4 Choice of data augmentation

We evaluate different data augmentations for SimCLR-Rank and SimSiam in Appendix A.2.7: the zeros augmentation with probabilities 0.1 and 0.7 and the gaussian augmentation with scales 1.0 and 2.0. We find that the gaussian augmentation with scale 1.0 works best for SimCLR-Rank and the zeroes augmentation with probability 0.1 works best for SimSiam.

## 5 Conclusion

We show that pretrained deep rankers can outperform GBDTs when the training data is low-signal across a variety of datasets, both public and proprietary. Our recommendation: practitioners should experiment with deep (pretrained) rankers in their own applications, because many real-world settings (like large-scale search or recommendations) exhibit sparsity or scarcity.

## 6 REPRODUCIBILITY

We provide a link to an anonymized repo with the code and commands needed to regenerate the results on the public datasets. We also provide experimental details in Section 4.3.2, Section 2.1, Section 4.1.2, Section 4.3.1, Appendix A.2.7, Section 4.1.1 which are paired along with the results on the public datasets. However, we do not provide full details for reproducibility on the results in Section 4.2, because the details are (1) proprietary and we are not legally allowed to release them (2) it would violate double-blind review anonymization.

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

# A    APPENDIX

## A.1    RELATED WORK

**Learning-To-Rank.** In our paper, we focus on the traditional LTR setting where the features are all numeric (tabular data). However, there is a line of work in LTR where raw text is also an input. In this case, one can leverage large language models in the ranking setting (Zhuang et al., 2023; Han et al., 2020; Yates et al., 2021; Nogueira et al., 2019; Mitra et al., 2018).

In tabular LTR problems, the dominant models currently used are gradient boosted decision trees (GBDTs) (Friedman, 2001), which are not deep learning models. GBDT models, which perform well on tabular data, are adapted to the LTR setting via losses that are surrogates for ranking metrics like NDCG. Surrogate losses (including LambdaRank/RankNet (Burges, 2010) and PiRank (Swezey et al., 2021)) are needed because many important ranking metrics (like NDCG) are non-differentiable. The combination of tree-based learners and ranking losses has become the de-facto standard in ranking problems, and deep models have yet to outperform them convincingly (Qin et al., 2021; Joachims, 2006; Ai et al., 2019; Bruch et al., 2019; Ai et al., 2018; Pang et al., 2020).

**Deep tabular models.** Given the success of neural methods in many other domains, there have been many attempts to adapt deep models to the tabular domain. TANGOS introduced special tabular-specific regularization to try to improve deep models' performance (Jeffares et al., 2023). FT-transformer and TabTransformer were introduced as transformer-based approaches to tabular data (Gorishniy et al., 2021; Huang et al., 2020). All these models have failed to convincingly outperform tree-based methods based on their own evaluations.

**Self-supervised learning.** Self-supervised learning (SSL) or unsupervised pretraining has improved performance in settings where there is a significant source of unlabeled data like text (Devlin et al., 2018) and images (Chen et al., 2020). In SSL, deep models are first pretrained on perturbed unlabeled data using self-supervised tasks to learn useful representations for the data. Then these models are finetuned for a downstream task with labeled data. Finetuning often takes one of two forms: (1) linear probing (a popular finetuning strategy in text and images (Chen & He, 2021; Chen et al., 2020; Peters et al., 2019)),where we freeze the pretrained model and only update the linear head during supervised finetuning, and (2) full finetuning, where we update the whole model during supervised finetuning (Devlin et al., 2018). Sometimes a mix of the two is used (Kumar et al., 2022). The core idea behind prominent SSL approaches like SimSiam and SimCLR is to carefully perturb input training samples, and train a representation that is consistent for transformations of the same sample. This provides robustness to natural perturbations and noise in data (Hendrycks et al., 2019).

Inspired by the success of pretraining and self-supervised learning in images and text, several works show how to apply SSL to unlabeled tabular data. One strategy is to corrupt tabular data and train a deep model to reconstruct it (Yoon et al., 2020; Majmundar et al., 2022; Ucar et al., 2021; Nam

et al., 2023b; Lin et al., 2023; Syed & Mirza, 2023; Hajiramezanali et al., 2022). Another approach is to use contrastive losses, which have been highly successful in the image domain (Chen et al., 2020). These methods are applicable because tabular data, like image data, is often composed of fixed-dimensional real vectors (Verma et al., 2021; Bahri et al., 2021; Lee & Shin, 2022; Hager et al., 2023; Liu et al., 2023; Darabi et al., 2021). Rubachev et al. (2022) evaluate a variety of different pretraining methods for tabular learning across many different datasets, finding that there is not a clear state of the art.

Our aim is to show that pretraining in tabular LTR can produce deep models that outperform GBDTs in ranking. To do so, we evaluate simple representative tabular pretraining methods from both the reconstructive and contrastive strategies: SimSiam (reconstructive) (Chen & He, 2021), SimCLR (contrastive) (Chen et al., 2020), SimCLR-Rank (ours, contrastive), and SubTab (reconstructive) (Ucar et al., 2021). **Note that our goal is not to show that any of these particular methods are necessarily the best, but rather that that pretraining in tabular LTR can produce deep models that outperform GBDTs in ranking.**

Finally, we summarize some related work on transfer learning in tabular learning. One direction revolves around pretraining models on common columns across many datasets (Zhu et al., 2023; Wang & Sun, 2022; Sui et al., 2023; Ye et al., 2023). Another direction leverage LLMs (large language models) to do few-shot tabular learning (Hegselmann et al., 2023; Liu et al., 2022; Nam et al., 2023a). These approaches are orthogonal to our goals in this work, which is not explicitly focused on transfer learning.

**Robustness in LTR.** There has been prior work on studying worst-case behavior (robustness) of rankers (Voorhees, 2005; Zhang et al., 2013; Goren et al., 2018; Wu et al., 2022b;a; Penha et al., 2022). Some previous metrics measure a model's robustness against adversarial attack (Goren et al., 2018; Wu et al., 2022a; 2023). Others measure the model's per-query performance variance on a dataset (Voorhees, 2005; Zhang et al., 2013; Wu et al., 2022b). Our outlier metric, Outlier-NDCG, is a departure from previous work because it is not directly a measure of robustness–it is possible for a model to perform *better* on outlier data.

## A.2 ADDITIONAL RESULTS

### A.2.1 OUTLIERS DETAILS

We systematically select outliers as follows: we generate a histogram with 100 bins for each feature across the *validation dataset*. For example, Istella (Lamkhede & Kofler, 2021) has 220 features, so we have 220 different histograms. For each histogram, we scan from left to right on the bins until we have encountered at least $G$ empty bins in a row, and if there is less than 1% of the validation set above this bin, then all the feature values above this bin are considered outliers. We also repeat this process right to left. Any test query group containing items with outlier feature values is labeled an outlier query group, and placed in the test outlier dataset.

Because different datasets have differently-sized typical gaps, we tune $G$ for each dataset (MSLR, Yahoo, Istella) such the resulting percentage of outlier queries is as close to 1% of the test set as we can get. MSLR has $G = 5$, with 0.65% (40/6072) outlier queries, Yahoo has $G = 20$, with 1.4% (30/2147) outlier queries, and Istella has $G = 32$, with 0.46% (34/7202) outlier queries. 1% is a hyperparameter that can be tuned according to the user's goals.

Here we present an example histogram of a feature in Istella_S.

### A.2.2 OMITTED TABLE ON RUNTIME COMPARISONS

Here we give the runtime comparisons between SimSiam, SimCLR, and SimCLR-Rank (our method).

### A.2.3 ADDITIONAL RESULTS ON LINEAR PROBING VS FINETUNING

Here we provide more results on linear probing vs finetuning, as discussed in Section 4.3.1. In Table 7 we give the results for different linear probing strategies when using SimSiam as the pretraining method. In Table 8 we give the results for different linear probing strategies when using

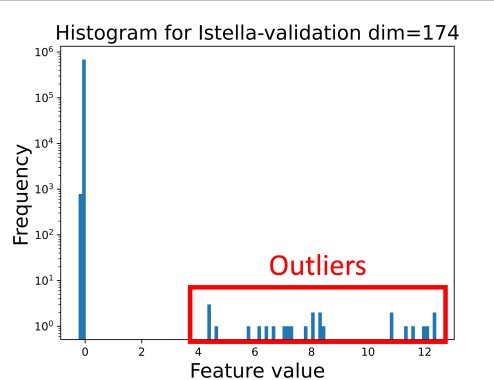

Figure 5: An example of outlier detection in Istella (Lucchese et al., 2016) for our Outlier-NDCG metric.

Table 6: Seconds per epoch comparison between pretraining methods. Average over 3 trials. The encoder we use for pretraining is the tabular ResNet (Gorishniy et al., 2021) with the final linear layer taken out.

| Method | MSLR | Yahoo Set1 | Istella |
|---|---|---|---|
| SimSiam | $0.8700 \pm 0.01$ | $0.130 \pm 0.00$ | $0.730 \pm 0.00$ |
| SimCLR | $102.81 \pm 1.14$ | $14.45 \pm 0.04$ | $69.26 \pm 0.16$ |
| SimCLR-Rank (Ours) | $9.4700 \pm 0.02$ | $1.970 \pm 0.01$ | $5.950 \pm 0.04$ |

SimCLR-Rank, with both the NDCG and Outlier-NDCG results (in the main paper we only include the NDCG results due to space constraints).

Table 7: SimSiam under different finetuning strategies on NDCG/Outlier-NDCG, averaged over 3 trials. We find that linear probing *and* MLP probing perform extremely poorly (except in Yahoo Set1, where MLP probing performs well).

| Method | MSLR (↑) | Yahoo Set1 (↑) | Istella (↑) |
|---|---|---|---|
| Linear probing | $0.2679 \pm 0.0007$ | $0.6089 \pm 0.0032$ | $0.3805 \pm 0.0042$ |
| Multilayer probing | $0.2764 \pm 0.0001$ | $\mathbf{0.6137 \pm 0.0022}$ | $0.4484 \pm 0.0020$ |
| Full finetuning | $\mathbf{0.3935 \pm 0.0034}$ | $0.6107 \pm 0.0035$ | $\mathbf{0.5618 \pm 0.0049}$ |
| | MSLR Outlier (↑) | Set1 Outlier (↑) | Istella Outlier (↑) |
| Linear probing | $0.1803 \pm 0.0033$ | $0.5088 \pm 0.002$ | $0.4407 \pm 0.0388$ |
| Multilayer probing | $0.1749 \pm 0.0023$ | $0.5157 \pm 0.0080$ | $0.5324 \pm 0.0002$ |
| Full finetuning | $\mathbf{0.3149 \pm 0.0119}$ | $\mathbf{0.52 \pm 0.0133}$ | $\mathbf{0.6348 \pm 0.0164}$ |

### A.2.4    ADDITIONAL RESULTS ON SIMCLR-RANK VS SIMSIAM

In this subsection we give the embedding plots comparing SimCLR-Rank, SimSiam, and fully supervised encoders on Yahoo Set1, where SimSiam performs the best. The interpretation is given in Section 4.3.3.

### A.2.5    ADDITIONAL RESULTS ON RELEVANCE SCORES

Here we provide the results on the relevance score setting for Outlier-NDCG in Figure 7. Result interpretation is given in Section 4.1.1.

Table 8: SimCLR-Rank under different finetuning strategies on NDCG/Outlier-NDCG, averaged over 3 trials. Multilayer probing performs moderately worse than full finetuning, and linear probing performs much worse than all other strategies.

| Method | MSLR (↑) | Yahoo Set1 (↑) | Istella (↑) |
|---|---|---|---|
| Linear probing | $0.3219 \pm 0.0224$ | $0.5202 \pm 0.0093$ | $0.4029 \pm 0.0073$ |
| Multilayer probing | $0.3890 \pm 0.0011$ | $0.5942 \pm 0.0016$ | $0.5813 \pm 0.0006$ |
| Full finetuning | $\mathbf{0.3959 \pm 0.0022}$ | $\mathbf{0.6022 \pm 0.0013}$ | $\mathbf{0.5839 \pm 0.0013}$ |
| | MSLR Outlier (↑) | Set1 Outlier (↑) | Istella Outlier (↑) |
| Linear probing | $0.2304 \pm 0.0332$ | $0.3811 \pm 0.0048$ | $0.4717 \pm 0.0166$ |
| Multilayer probing | $\mathbf{0.2969 \pm 0.0009}$ | $0.4888 \pm 0.0046$ | $\mathbf{0.6369 \pm 0.0045}$ |
| Full finetuning | $0.2892 \pm 0.0025$ | $\mathbf{0.5143 \pm 0.0055}$ | $0.6352 \pm 0.0140$ |

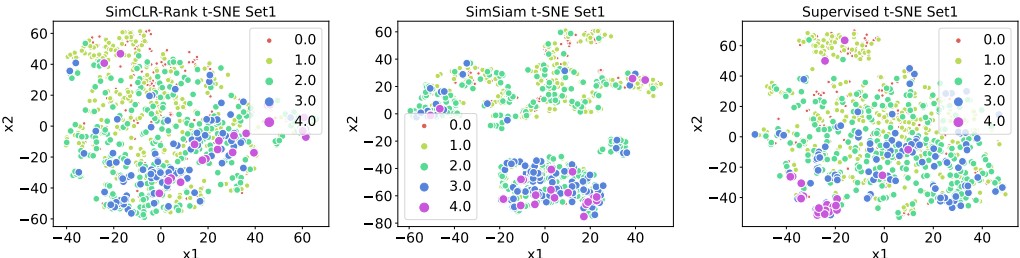

Figure 6: We plot the t-SNE plots of embeddings produced by three different encoders: (1) pretrained by SimCLR-Rank, (2) pretrained by SimSiam, (3) trained via supervised training on the entire training set on roughly 1000 samples from Yahoo Set1. We find that SimSiam's more aggressive clustering sorts embeddings fairly well on Yahoo Set1.

### A.2.6 ADDITIONAL RESULTS ON BINARY LABEL GENERATION

In this part of the appendix we provide additional results on binary label generation as detailed in Section 4.1.2. In Table 10 we give the dataset statistics when we use $\tau_{\text{target}} = 4.25$, and in Table 12 we give the dataset statistics when we use $\tau_{\text{target}} = 5.1$ In Table 11 we give the comparison between pretrained and non-pretrained rankers following the methodology in Section 4.1.2 and find that GBDTs perform the best there (under not-very-sparse binary label conditions), while in Table 13 we find that pretrained rankers perform the best under sparse binary label conditions.

### A.2.7 ADDITIONAL RESULTS ON DATA AUGMENTATION

In Table 14 we show the performance of each augmentation choice on SimCLR-Rank and in Table 15 we show the performance of each augmentation choice. We use the methodology described in Section 4.1.1, and have 0.2% of training QGs labeled.

Table 9: Train and validation dataset statistics on the MSLRWEB30K, Yahoo Set1, and Istella_S datasets.

| Dataset | # query groups | | | # features |
|---|---|---|---|---|
| | train | val | test | - |
| MSLRWEB30K | 18151 | 6072 | 6072 | 136 |
| Yahoo Set1 | 14477 | 2147 | 5089 | 700 |
| Istella_S | 19200 | 7202 | 6523 | 220 |

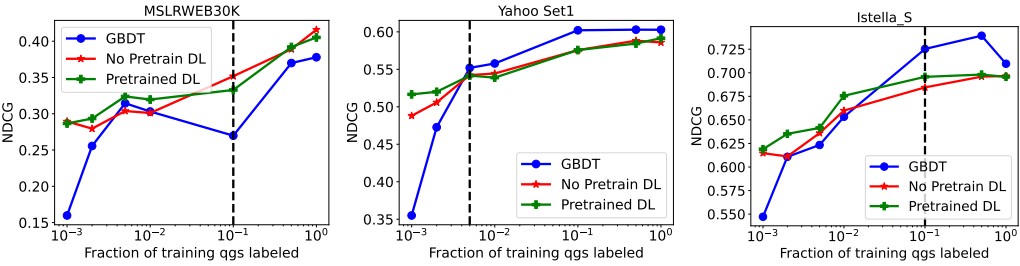

Figure 7: We compare Outlier-NDCG ($\uparrow$) pretrained rankers, non-pretrained DL rankers, and GBDT rankers as we change the percentage of training query groups that are labeled. To the left of the black dotted line, pretrained rankers perform the best. Points are averages over three trials. To the left of the black dotted vertical line, pretrained rankers are (1) significantly better on outliers than GBDTs at the $p = 0.05$ level using a one-sided $t$-test, and (2) on average better on outliers than all other non-pretrained methods.

Table 10: Train and validation dataset statistics on MSLRWEB30K, Yahoo Set1, and Istella_S after binary label generation with $\tau_{\text{target}} = 4.25$. The test set is left untouched. Labeled QGs are those query groups that have at least one item with $y = 1$.

| Dataset | # query groups | | # labeled QGs | | # items per QG | | # positives per QG | |
|---|---|---|---|---|---|---|---|---|
| | train | val | train | val | train | val | train | val |
| MSLRWEB30K | 18151 | 6072 | 15.9% | 16.2% | 124.35 | 122.38 | 1.4% | 1.4% |
| Yahoo Set1 | 14477 | 2147 | 11.1% | 10.8% | 30.02 | 30.32 | 6.2% | 6.0% |
| Istella_S | 19200 | 7202 | 49.0% | 49.5% | 106.36 | 94.98 | 1.4% | 1.6% |

Table 11: We compare pretrained models to non-pretrained models in the binary label setting with $\tau_{\text{target}} = 4.25$ (Section 4.1.2) on NDCG averaged over three trials. We follow the methodology in Section 4.1.2.

| Method | MSLR ($\uparrow$) | Yahoo Set1 ($\uparrow$) | Istella ($\uparrow$) |
|---|---|---|---|
| GBDT | $0.3616 \pm 0.0000$ | $0.6233 \pm 0.0000$ | $0.6251 \pm 0.0000$ |
| No Pretrain DL | $0.3552 \pm 0.0015$ | $0.6297 \pm 0.0005$ | $0.6147 \pm 0.0006$ |
| Pretrained DL | $0.3602 \pm 0.0007$ | $0.6272 \pm 0.0024$ | $0.6243 \pm 0.0007$ |
| | MSLR Outlier ($\uparrow$) | Set1 Outlier ($\uparrow$) | Istella Outlier ($\uparrow$) |
| GBDT | $0.2792 \pm 0.0000$ | $0.5352 \pm 0.0000$ | $0.7393 \pm 0.0000$ |
| No Pretrain DL | $0.3142 \pm 0.0063$ | $0.5389 \pm 0.0036$ | $0.6631 \pm 0.0102$ |
| Pretrained DL | $0.2923 \pm 0.0079$ | $0.5471 \pm 0.0050$ | $0.6945 \pm 0.0041$ |

Table 12: Train and validation dataset statistics on MSLRWEB30K, Yahoo Set1, and Istella_S after binary label generation with $\tau_{\text{target}} = 5.1$. The test set is left untouched. Labeled QGs are those query groups that have at least one item with $y = 1$.

| Dataset | # query groups | | # labeled QGs | | # items per QG | | # positives per QG | |
|---------|------|------|-------|------|--------|--------|-------|------|
| | train | val | train | val | train | val | train | val |
| MSLRWEB30K | 18151 | 6072 | 1.1% | 1.3% | 124.35 | 122.38 | 0/8% | 0.7% |
| Yahoo Set1 | 14477 | 2147 | 0.6% | 0.6% | 30.02 | 30.32 | 5.4% | 5.1% |
| Istella_S | 19200 | 7202 | 3.0% | 2.7% | 106.36 | 94.98 | 1.1% | 1.3% |

Table 13: We compare pretrained models to non-pretrained models in the binary label setting with $\tau_{\text{target}} = 5.1$ (Section 4.1.2) on NDCG averaged over three trials. We follow the methodology in Section 4.1.2. ♣ indicates metrics on which pretrained rankers outperform non-pretrained rankers significantly via a $p < 0.05$ t-test.

| Method | MSLR (↑) | Yahoo Set1 (↑) | Istella (↑) |
|--------|----------|----------------|-------------|
| GBDT | $0.2844 \pm 0.0000$ | $0.5782 \pm 0.0000$ | $0.5638 \pm 0.0000$ |
| No Pretrain DL | $0.3432 \pm 0.0065$ | $0.5703 \pm 0.0246$ | $0.5722 \pm 0.0080$ |
| Pretrained DL | $\mathbf{0.3564 \pm 0.0054}$♣ | $\mathbf{0.6031 \pm 0.0064}$ ♣ | $\mathbf{0.599 \pm 0.0014}$ ♣ |
| | MSLR Outlier (↑) | Set1 Outlier (↑) | Istella Outlier (↑) |
| GBDT | $0.2416 \pm 0.0000$ | $0.5011 \pm 0.0000$ | $0.6640 \pm 0.0000$ |
| No Pretrain DL | $0.2441 \pm 0.0089$ | $0.4593 \pm 0.0317$ | $0.6334 \pm 0.0054$ |
| Pretrained DL | $\mathbf{0.2553 \pm 0.0161}$ | $\mathbf{0.54 \pm 0.0055}$♣ | $\mathbf{0.6730 \pm 0.0090}$ ♣ |

Table 14: We compare different augmentation strategies for SimCLR-Rank. The methodology used is the one in Section 4.1.1, with 0.2% of training QGs labeled.

| Augmentation | MSLR (↑) | Yahoo Set1 (↑) | Istella (↑) |
|--------------|----------|----------------|-------------|
| Zeros p=0.1 | $0.3830 \pm 0.0007$ | $\mathbf{0.6022 \pm 0.0013}$ | $0.5820 \pm 0.0024$ |
| Zeros p=0.7 | $0.3737 \pm 0.0025$ | $0.5998 \pm 0.0053$ | $0.5646 \pm 0.0048$ |
| Gaussian scale=1.0 | $\mathbf{0.3959 \pm 0.0022}$ | $0.5998 \pm 0.0026$ | $\mathbf{0.5839 \pm 0.0013}$ |
| Gaussian scale=2.0 | $0.3907 \pm 0.0025$ | $0.5953 \pm 0.0043$ | $0.5809 \pm 0.0010$ |
| Augmentation | MSLR Outlier (↑) | Set1 Outlier (↑) | Istella Outlier (↑) |
| Zeros p=0.1 | $0.2782 \pm 0.0036$ | $\mathbf{0.5143 \pm 0.0055}$ | $\mathbf{0.6408 \pm 0.0063}$ |
| Zeros p=0.7 | $0.2730 \pm 0.0054$ | $0.5062 \pm 0.0070$ | $0.6141 \pm 0.0064$ |
| Gaussian scale=1.0 | $\mathbf{0.2892 \pm 0.0025}$ | $0.4963 \pm 0.0024$ | $0.6352 \pm 0.0140$ |
| Gaussian scale=2.0 | $0.2886 \pm 0.0096$ | $0.4875 \pm 0.0054$ | $0.6327 \pm 0.0191$ |

Table 15: We compare different augmentation strategies for SimSiam. The methodology used is the one in Section 4.1.1, with 0.2% of training QGs labeled.

| Augmentation | MSLR (↑) | Yahoo Set1 (↑) | Istella (↑) |
|--------------|----------|----------------|-------------|
| Zeros p=0.1 | $\mathbf{0.3935 \pm 0.0034}$ | $\mathbf{0.6107 \pm 0.0035}$ | $0.5618 \pm 0.0049$ |
| Zeros p=0.7 | $0.3911 \pm 0.0003$ | $0.6076 \pm 0.0072$ | $\mathbf{0.5660 \pm 0.0047}$ |
| Gaussian scale=1.0 | $0.3782 \pm 0.0093$ | $0.6010 \pm 0.0026$ | $0.5612 \pm 0.0060$ |
| Gaussian scale=2.0 | $0.3860 \pm 0.0011$ | $0.6100 \pm 0.0089$ | $0.5587 \pm 0.0047$ |
| Augmentation | MSLR Outlier (↑) | Set1 Outlier (↑) | Istella Outlier (↑) |
| Zeros p=0.1 | $\mathbf{0.3149 \pm 0.0119}$ | $\mathbf{0.5200 \pm 0.0133}$ | $\mathbf{0.6348 \pm 0.0164}$ |
| Zeros p=0.7 | $0.3002 \pm 0.0060$ | $0.5081 \pm 0.0097$ | $0.6201 \pm 0.0104$ |
| Gaussian scale=1.0 | $0.2585 \pm 0.0371$ | $0.4893 \pm 0.0084$ | $0.6145 \pm 0.0160$ |
| Gaussian scale=2.0 | $0.2929 \pm 0.0021$ | $0.5163 \pm 0.0101$ | $0.5905 \pm 0.0146$ |

