# OpenReview forum: "Pretrained deep models outperform GBDTs in Learning-To-Rank under label scarcity"
_ICLR.cc/2024/Conference — Submitted to ICLR 2024_

### Official Review · Reviewer_i3NM · 2023-10-27

**Soundness:** 3 good
**Presentation:** 3 good
**Contribution:** 3 good
**Rating:** 6
**Confidence:** 4

**Summary:**

The paper studies pre-trained DNN for tabular LTR problems, which is an underexplored problem. The major contribution of the paper is to identify areas where this might help, including query sparsity and label sparisity scenarios. Technically the paper proposes a loss function specific to ranking data nature based on existing contrastive learning objectives. Experiments are mainly compared against GBDT and DNN without pre-training, and it’s shown that there’re some benefits in the discussed scenarios. Experiments are also conducted offline on a private industry dataset.

**Strengths:**

S1: To the reviewer’s knowledge, this is the first work that shows some promises for pre-trained DNNs for the LTR task. The reviewer thought about the direction but It was not intuitively clear how to do it or if it has benefits. The paper still has several caveats but is a decent exploration in some aspects.

S2: the motivation of the ranking contrastive loss is clear and easy to understand - it is clear what the hard negatives are for LTR problems, so it is good to leverage that.

S3: It is good to identify two scenarios where pre-trained deep models might help, including query sparse and label sparse scenarios.

**Weaknesses:**

W1: change over SimCLR is incremental - the major weakness of SimCLR for non-ranking problems was complexity. The authors made a good point that hard negatives are clear for LTR (mentioned in S1), but SimCLR using small batches will likely largely resolve the issues? So the necessity for a new loss is not very convincing - in fact, the authors did not comprehensively compare with that baseline. - also, sometimes having easy negatives may improve the generalization of learning - this may need deeper study. SimSiam does not look to be a competitive baseline considering the factors. So the proposal of the new loss is not very convincing.

W2: extendability. The reviewer thought about this problem before and one major difference between tabular vs text is, tabular datasets assume a feature space, while text encoders are for text in general that is easy to generalize. E.g., one can easily add more text fields and everything is still kind of in the “text space”, but adding new features to the tabular table may completely invalidate the pre-training in the previous feature space. So when the feature space changes, which can be common in practice, the pre-training needs to be done again. Also, the pre-training needs to be done for each dataset separately, unlike text domains where one model may be sufficient. Thus pre-trained models on tabular datasets may be much limiting in general.

W3: It is unclear why no online results are provided for the industry dataset. Overall it is not clear how significant the results are in this section due to unclear baselines, private dataset, and unclear real-world implications.

**Questions:**

Please discuss the weaknesses listed above.
Typo in figure2 caption? non-pretrained counterparts?

---

> ### Author Response · Authors · 2023-11-19
> **Reply part 1**
>
> Thank you for your review and feedback. We would like to first take an opportunity to clarify the impact and contributions of our work.
>
> We strongly believe our work will significantly impact industries that work with LTR problems (search, recommendation), by causing them to revise, or at least revisit, current SOTA methods. Today many large companies use GBDTs for ranking tabular data, and this approach is SOTA even in the research community. The novel starting point of our work is the often ignored fact that most of the data collected in LTR systems is unlabeled. We have shown through extensive experiments on standard public datasets and a large-scale private dataset that pretrained deep rankers can leverage unlabeled data to consistently outperform GBDTs, with outsized gains in robustness towards outlier data.
>
> All reviewers agree we have convincingly demonstrated this. Reviewer 1XYq: “Experimentally, authors show that pretrained models can outperform GBDTs, which is one of the strongest baselines, under label scarcity setting.” Reviewer oW1i: “the paper highlights the superior performance of pretrained deep rankers, especially on outlier queries, in scenarios with limited labeled data.” Reviewer i3NM: “The major contribution of the paper is to identify areas where [pretrained deep rankers] help, including query sparsity and label sparisity scenarios.”
>
> We now address the concerns of the reviewer.
>
> **Will decreasing the number of negatives in SimCLR be sufficient to apply SimCLR to ranking problems?**
>
> Thank you for asking this insightful question. First, we would like to clarify that our primary goal is to demonstrate that DL rankers built using pre-training techniques can achieve SOTA performance in LTR settings, and consequently beat GBDT models (prior SOTA). Only as a secondary goal, we investigate a few simple but effective pre-training strategies such as SimSiam, SimCLR, and our variant SimCLR-Rank. While we identified SimCLR-Rank has a positive impact in a wide variety of LTR scenarios, we also show that SimSiam works better for our large-scale proprietary dataset. Our goal is not to show that any one particular method is the best for pretraining in LTR.
>
> As the reviewer correctly identified, SimCLR-Rank is different from SimCLR in two aspects: (1) it uses items from the same query group as freely available hard-negatives, (2) it reduces the number of negatives from batchsize to at most query group size and thus allow better computational scaling with batchsize. To identify which of these contribute more towards SimCLR-Rank’s performance, we performed an additional experiment below (as suggested by the reviewer), comparing SimCLR-Rank with a variant of SimCLR (which we call as SimCLR-sample). In SimCLR-sample, instead of using all the other items as negatives, we uniformly sample a constant number of negatives from the batch. This reduces the computational complexity from quadratic in batchsize (of SimCLR) to linear in batchsize. In the below experiments we follow the setup in Subsection 4.3.2 and SimCLR-Rank and SimCLR-sample use the same number of negatives and batchsize.
>
> We find that SimCLR-Rank performs better on MSLR/Istella, while SimCLR-sample performs better on Yahoo. We note that MSLR/Istella are much sparser than Yahoo (see Table 12 in the paper), and are more representative of typical search and recommendations applications. Therefore we still believe that SimCLR-Rank which selects hard-negatives from the same query group is a useful pre-training method for the LTR toolbox and can be highly effective in many situations.
>
> | Method | MSLR | Yahoo | Istella |
> | --- | ----------- | - | - |
> | SimCLR-Rank| **0.3929 $\pm$ 0.0018** | 0.5989 $\pm$ 0.0010 | **0.5830 $\pm$ 0.0013** |
> | SimCLR-sample | 0.3890 $\pm$ 0.0008 | **0.6056 $\pm$ 0.0047** | 0.5787 $\pm$ 0.0035 |
>
> **Pretrained models in tabular datasets are not very extendable.**
>
> The reviewer’s observation that our pretrained models cannot take advantage of transfer learning is correct. Tabular datasets have specific features which can be very different. In fact, each tabular dataset is its own domain. Because in general there are no common features between tabular datasets, it is not clear how to have common knowledge. Levin et al. [1] show that if features are shared, it is possible to transfer knowledge between datasets, but in general it is not known how to do transfer learning in tabular data. The same shortcoming is true of GBDTs. Transfer learning for tabular data is a question of independent interest, and beyond the scope of our paper.

---

> ### Author Response · Authors · 2023-11-19
> **Reply part 2**
>
> **It is not clear how significant the results are in the private dataset section.**
>
> We agree with the reviewer that only online experiments can reveal the true impact of an LTR model due to the counterfactual nature of the LTR problem. That is from offline experiments, it is not clear how the user might have responded to the ranking provided by the model. However  running online experiments on real-world user traffic needs to be carefully considered as it has a potential opportunity cost. Unfortunately, due to superseding business priorities we could not run online experiments. Similarly, unfortunately at this point we cannot reveal more details of the baseline, but we will try our best to provide more details in the camera ready version.
>
> To provide more confidence in our private dataset results, we updated our Table 3 to provide error bars to our results.
>
> | Target | Method | Delta% NDCG | Delta% Outlier NDCG |
> | --- | ----------- | - | - |
> | Purchase| Baseline | +0.00% | +0.00% |
> |  | SimSiam | **+1.75% $\pm$ 0.13%** | **+29.66% $\pm$ 0.84%**|
> | | SimCLR-Rank | +0.18% $\pm$ 0.13% | +2.19% $\pm$ 0.71% |
> | Relevance| Baseline | +0.00% | +0.00% |
> |  | SimSiam | **+2.78% $\pm$ 0.06%** | **+26.88% $\pm$ 0.35%**|
> | | SimCLR-Rank | +0.85% $\pm$ 0.06% | +2.99% $\pm$ 0.31% |
>
> Tabular (a.k.a.numerical) ML (e.g. LTR) systems are important for modern enterprises and therefore validating research hypotheses on large-scale practical datasets is highly valuable. Unfortunately, this is challenging due to the expensive and proprietary nature of such data and most LTR papers [2,3,4] do not prove their hypotheses on practical large-scale datasets. We demonstrate pretraining rankers achieve SOTA on public datasets but also emphatically on a large-scale commercial dataset. Therefore, we believe our results will be useful to the LTR community and the wider Tabular-DL community.
>
> We thank the reviewer for their consideration, and are happy to help with any further concerns. If we have addressed the concerns we would appreciate it if the reviewer could consider raising their score.
>
> [1] Transfer Learning with Deep Tabular Models. Roman Levin, Valeriia Cherepanova, Avi Schwarzschild, Arpit Bansal, C. Bayan Bruss, Tom Goldstein, Andrew Gordon Wilson, Micah Goldblum. ICLR 2023.
> [2] Are Neural Rankers still Outperformed by Gradient Boosted Decision Trees?, Qin et al., ICLR 2021.
> [3] Learning Groupwise Multivariate Scoring Functions Using Deep Neural Networks., Ai et al., SIGIR 2019.
> [4] SetRank: Learning a Permutation-Invariant Ranking Model for Information Retrieval., Pang et al., SIGIR 2020

---

> ### Author Response · Authors · 2023-11-21
> **Follow-up to replies, additional info**
>
> We thank the reviewer for their helpful feedback during this rebuttal process.
>
> A follow-up W1: we produced a comprehensive comparison between SimCLR-Rank, SimSiam, and many SOTA tabular pretraining baselines in a reply to another reviewer: https://openreview.net/forum?id=Dk1ybhMrJv&noteId=1Z4qMYixVv. We hope this, combined with our earlier responses, gives the reviewer more confidence in our results.
>
> We thank the reviewer for their time. Please kindly let us know if there are additional comments you have for us.

---

> ### Author Response · Authors · 2023-11-22
> **Gentle request for final feedback**
>
> We would like to thank the reviewer again for their encouraging feedback. While we have an opportunity, we would like to make an additional point about **W2 (extendability)**:
>
> It is common for unsupervised pretraining (self-supervised learning) papers to not consider transferability (including in very high-impact and influential works). Examples include: tabular classification [1,2,3,7], image classification [4,5,6], and text classification [4,5]. Like these papers, we show that pretraining in the same dataset provides significant benefits, and transfer learning is not needed to achieve these results.
>
> Extendability is even harder to prove in LTR than the general tabular setting. In tabular data (where transferability is hard, as noted in Ucar et al. [1]), overlapping columns seems to be what is needed to achieve transferability [8]. Getting overlapping columns seems to be nearly impossible in the three benchmark LTR public datasets, two of which (Yahoo, Istella) [9,10] do not specify what the columns even mean (due to the need to protect business trade secrets). The most transparent, MSLRWEB30K [11], also has many columns whose meanings may not make sense outside of the context of the dataset (for example, features that are the outputs of Microsoft internal models).
>
> We hope our discussion has adequately addressed your feedback on (1) SimCLR-Rank's utility, (2) extendability of results, (3) the significance of the results. Please kindly let us know if there are additional comments you have for us.
>
>
>
> [1] SubTab: Subsetting Features of Tabular Data for Self-Supervised Representation Learning. Talip Ucar, Ehsan Hajiramezanali, Lindsay Edwards. NeurIPS 2021.
> [2] VIME: Extending the Success of Self- and Semi-supervised Learning to Tabular Domain. Jinsung Yoon, Yao Zhang, James Jordon, Mihaela van der Schaar. NeurIPS 2020.
> [3] Towards Domain-Agnostic Contrastive Learning. Vikas Verma, Minh-Thang Luong, Kenji Kawaguchi, Hieu Pham, Quoc V. Le. ICML 2021.
> [4] Contrastive learning with hard negative samples. Joshua Robinson, Ching-Yao Chuang, Suvrit Sra, Stefanie Jegelka. ICLR 2021.
> [5] Debiased Contrastive Learning. Ching-Yao Chuang, Joshua Robinson, Lin Yen-Chen, Antonio Torralba, Stefanie Jegelka. NeurIPS 2020.
> [6] Robust Contrastive Learning Using Negative Samples with Diminished Semantics. Songwei Ge, Shlok Mishra, Chun-Liang Li, Haohan Wang, David Jacobs. NeurIPS 2021.
> [7] SCARF: Self-Supervised Contrastive Learning using Random Feature Corruption. Dara Bahri, Heinrich Jiang, Yi Tay, Donald Metzler. ICLR 2022.
> [8] Transfer Learning with Deep Tabular Models. Roman Levin, Valeriia Cherepanova, Avi Schwarzschild, Arpit Bansal, C. Bayan Bruss, Tom Goldstein, Andrew Gordon Wilson, Micah Goldblum. ICLR 2023.
> [9] Yahoo! Learning to Rank Challenge Overview. Olivier Chapelle, Yi Chang.
> [10] Fast Ranking with Additive Ensembles of Oblivious and Non-Oblivious Regression Trees. Domenico Dato et al.
> [11] Introducing LETOR 4.0 Datasets. Tao Qin, Tie-Yan Liu.

---

> > ### Author Response · Authors · 2023-11-22
> > **Gentle reminder of the discussion period's last day**
> >
> > We thank you for your time and consideration. Your review has been very helpful in improving our paper. Today is the end of the discussion period, and we are eagerly waiting to hear any additional feedback you might still have about the paper.
> >
> > We hope that we have addressed your concerns on (1) SimCLR-Rank, (2) Extendability, and (3) significance. Please let us know if there’s anything more we can do to improve our paper and/or increase our score.

---

### Official Review · Reviewer_oW1i · 2023-10-30

**Soundness:** 3 good
**Presentation:** 3 good
**Contribution:** 2 fair
**Rating:** 5
**Confidence:** 5

**Summary:**

This paper investigates the application of unsupervised pretraining to deep learning models for Tabular Learning-To-Rank (LTR) problems and demonstrates consistent outperformance of Gradient Boosted Decision Trees (GBDTs) and other non-pretrained rankers when there is more unlabeled data than labeled data. They introduce LTR-specific pretraining strategies, including the SimCLR-Rank loss, and show significant improvements in NDCG. Additionally, the paper highlights the superior performance of pretrained deep rankers, especially on outlier queries, in scenarios with limited labeled data.

**Strengths:**

1. This article is characterized by clear and comprehensible writing, presenting methods that are straightforward and easily implementable.

2. Through experimentation, this paper demonstrates that pre-trained deep models can achieve performance levels close to, or even surpass, GBDT in ranking tasks. This discovery holds practical value.

3. The paper introduces a pre-training approach that leverages the nature of learning to rank problems, demonstrating reasonable effectiveness, and in certain scenarios, outperforming simclr.

**Weaknesses:**

1. This paper exhibits notable deficiencies in the aspects of experimental comparisons and discussions on related work. The experimental comparison methodology only considers comparisons between fine-tuning or probing methods based on pre-trained models, as well as MLP models. I believe that in the realm of learning to rank and tabular data, there are likely more recent deep learning methods that could serve as baselines for comparison. Proper discussions about these methods should also be incorporated into the related work section. Currently, I find the discussion on related work to be inadequate. This would aid in addressing certain evident issues, such as whether current methods for using deep learning to learn tabular data also employ contrastive learning for pre-training and what distinguishes them from the approach presented in this paper.

2. Regarding the methodology, apart from applying SimCLR and SimSiam directly to pre-training for ranking tasks, the primary contribution of this paper is the SimCLR-Rank method. The key difference between this method and SimCLR lies in narrowing down the computation of contrastive learning from the entire batch to a single QG. This method possesses a degree of rationality and can effectively exploit the characteristics of the learning to rank task. However, in essence, this approach distills the semantic information contained in QG itself into the pre-trained model, similar to knowledge distillation (typically, QG is obtained through some form of retrieval, and the retrieval model can be viewed as a strong "teacher" model, while the pre-training model in this paper serves as a "student" model). Based on this observation, I believe that if the features from the retrieval model are input into GBDT, it could potentially achieve better performance, which is easily attainable in practical industrial applications. If this were the case, the application prospects of the method proposed in this paper would become rather limited.

3. In SimCLR, dissimilar embeddings correspond to smaller weights, while similar embeddings correspond to larger weights, indicating its adaptive ability to mine hard samples. In the learning to rank scenario, samples from the same QG should be relatively similar, and samples from different QGs should be dissimilar. This implies that the original SimCLR method, even when calculating softmax over the entire batch, primarily emphasizes the samples from the same QG, aligning with the role of SimCLR-Rank. Therefore, I believe that the potential improvement that SimCLR-Rank can bring to typical ranking situations may be somewhat limited. I hope the authors can provide further explanations regarding this issue.

**Questions:**

Please refer to Weaknesses.

---

> ### Author Response · Authors · 2023-11-19
> **Reply part 1**
>
> Thank you for your review and feedback.  We would like to first take an opportunity to clarify the impact and contributions of our work.
>
> We strongly believe our work will significantly impact industries that work with LTR problems (search, recommendation), by causing them to revise, or at least revisit, current SOTA methods. Today many large companies use GBDTs for ranking tabular data, and this approach is SOTA even in the research community. The novel starting point of our work is the often ignored fact that most of the data collected in LTR systems is unlabeled. We have shown through extensive experiments on standard public datasets and a large-scale private dataset that pretrained deep rankers can leverage unlabeled data to consistently outperform GBDTs, with outsized gains in robustness towards outlier data.
>
> All reviewers agree we have convincingly demonstrated this. Reviewer 1XYq: “Experimentally, authors show that pretrained models can outperform GBDTs, which is one of the strongest baselines, under label scarcity setting.” Reviewer oW1i: “the paper highlights the superior performance of pretrained deep rankers, especially on outlier queries, in scenarios with limited labeled data.” Reviewer i3NM: “The major contribution of the paper is to identify areas where [pretrained deep rankers] help, including query sparsity and label sparisity scenarios.”
>
> We now address the concerns of the reviewer.
>
> **The paper is missing important comparisons with other tabular SSL methods.**
>
> We thank the reviewer for bringing this to our attention. We added a comprehensive and expanded related works section to Appendix A.1 (page 14 in the appendix, colored in blue) discussing the current state of pretraining for tabular (a.k.a numerical) dataset. After the paper decision, we plan to reorder the paper to place it into the main text.
>
> First, we would like to clarify that our primary goal is to demonstrate that DL rankers built using pre-training techniques can achieve SOTA performance in LTR settings, and consequently beat GBDT models (prior SOTA). Only as a secondary goal, we investigate a few simple but effective pre-training strategies such as SimSiam, SimCLR, and our variant SimCLR-Rank. While we identified SimCLR-Rank has a positive impact in a wide variety of LTR scenarios, we also show that SimSiam works better for our large-scale proprietary dataset.  Our goal is not to show that any one particular method is the best for pretraining in LTR.
>
> SimCLR-Rank is different from SimCLR in two aspects: (1) it uses items from the same query group as freely available hard-negatives, (2) it reduces the number of negatives from batchsize to at most query group size and thus allow better computational scaling with batchsize. To identify which of these contribute more towards SimCLR-Rank’s performance, we performed an additional experiment below (as suggested by Reviewer i3NM), comparing SimCLR-Rank with a variant of SimCLR (which name SimCLR-sample). In SimCLR-sample, instead of using all the other items as negatives, we uniformly sample a constant number of negatives from the batch. This reduces the computational complexity from quadratic in batchsize (of SimCLR) to linear in batchsize. In the below experiments we follow the setup in Subsection 4.3.2 and SimCLR-Rank and SimCLR-sample use the same number of negatives and batchsize.
>
> | Method | MSLR | Yahoo | Istella |
> | --- | ----------- | - | - |
> | SimCLR-Rank| **0.3929 $\pm$ 0.0018** | 0.5989 $\pm$ 0.0010 | **0.5830 $\pm$ 0.0013** |
> | SimCLR-sample | 0.3890 $\pm$ 0.0008 | **0.6056 $\pm$ 0.0047** | 0.5787 $\pm$ 0.0035 |
>
> As an additional comparison (suggested by reviewer 1XYq), we also evaluate SubTab [1] in the setting of Subsection 4.3.2.  SubTab’s pretraining objective divides input features into multiple subsets (in the language of computer vision, “multiple views”) and trains an autoencoder to reconstruct the original input features.  To pretrain using SubTab, we divide the input features into 4 subsets with 75% overlap, and with input corruptions of 15% masking probability and Gaussian noise of scale 0.1 (suggested in the SubTab paper as a good setting).  The finetuning strategy and the choice for encoder model is the same as for SimCLR-Rank and SimCLR-sample.  We find that SubTab does very poorly on the MSLR and Istella datasets while performing respectably on Yahoo (though worse than both SimCLR-Rank and SimCLR-Sample).
>
> | Method | MSLR | Yahoo | Istella |
> | --- | ----------- | - | - |
> | SimCLR-Rank| **0.3929 $\pm$ 0.0018** | 0.5989 $\pm$ 0.0010 | **0.5830 $\pm$ 0.0013** |
> | SimCLR-sample | 0.3890 $\pm$ 0.0008 | **0.6056 $\pm$ 0.0047** | 0.5787 $\pm$ 0.0035 |
> |SubTab | 0.2879 $\pm$ 0.0019 | 0.5889 $\pm$ 0.0021 | 0.4700 $\pm$ 0.0031 |

---

> ### Author Response · Authors · 2023-11-19
> **Reply part 2**
>
> **Could we use features from the retrieval model to improve GBDT performance over SimCLR-Rank?**
>
> We thank the reviewer for this very interesting conjecture.
>
> First, we would like to note that the features of the retrieval model are in fact used in our LTR experiments on the private dataset. Despite this, SimCLR-Rank still outperforms the baseline model. Second, in the public datasets we do not have access to the upstream retrieval model features to test this hypothesis.
>
> The last observation we would like to make is that retrieval models are typically not large or powerful in applications like search or recommendations. They are typically designed to be fast models (e.g. KNNs or even handcrafted rules) with high recall (and potentially low precision) to quickly filter millions or billions of items into hundreds of possibly relevant items. Since they are trained for recall and not for precision, these models and their features may find it difficult to differentiate and rank the relevant items in the query group. Therefore, we conjecture that  distilling from retrieval to rankers might not work well.

---

> ### Author Response · Authors · 2023-11-19
> **Reply part 3**
>
> **The potential improvement of SimCLR-Rank over SimCLR might be limited in practical situations.**
>
> We thank the reviewer for their insightful question.  As the reviewer correctly identified, all the negative samples in the SimCLR-Rank loss are present in the SimCLR loss as well, so the qualitative and quantitative behaviors of the two algorithms are likely similar.
>
> However, SimCLR is difficult or impossible to use in large-scale big-data settings because of its high runtime and space complexity (which is quadratic in the batch size).  Reviewer i3NM also noted that SimCLR has this same shortcoming: “the major weakness of SimCLR for non-ranking problems was complexity.”
>
> To overcome this issue, we leverage the fact that in LTR the hard negatives are in the same query group (as the reviewer correctly notices), so we do not have to use the entire batch as negatives.  This (1) provides orders of magnitude improvement in speed (Table 6 in the paper, page 16), and (2) performs slightly better than SimCLR even with the runtime and memory savings (Table 5 in the paper, page 9).
>
> To summarize, we do not claim that SimCLR-Rank’s loss will produce significantly different outcomes than SimCLR.  However, SimCLR-Rank is significantly faster, which makes it applicable in real-world big data settings for ranking.
>
> We thank the reviewer for their consideration, and are happy to help with any further concerns. If we have addressed the concerns we would appreciate it if the reviewer could consider raising their score.

---

> ### Author Response · Authors · 2023-11-19
> **Reply part 4**
>
> [1] SubTab: Subsetting Features of Tabular Data for Self-Supervised Representation Learning. Talip Ucar, Ehsan Hajiramezanali, Lindsay Edwards. NeurIPS 2019.

---

> > ### Comment · Reviewer_oW1i · 2023-11-19
> >
> > I appreciate the author's diligent efforts in addressing the issues I raised. While I acknowledge that the author's responses offer resolutions to some of the concerns, addressing certain issues, such as the comparative analysis with existing tabular data methodologies, poses a formidable challenge within the constraints of time. Presently, the author has introduced only a singular comparative method, a move that evidently lacks compelling strength.
> >
> > In terms of the novelty of the paper, the author concedes that "we do not claim that SimCLR-Rank’s loss will produce significantly different outcomes." Consequently, it becomes challenging to attribute a substantial level of innovation to this work. The exploration of self-supervised learning in the context of tabular data is intriguing; however, confining the scope of this paper solely to its application in the niche domain of learning-to-rank significantly diminishes its impact as a comprehensive experimental analysis.
> >
> > While I am inclined to marginally elevate the evaluation scores, I maintain the stance that substantial revisions are imperative for this manuscript to align with the esteemed standards of ICLR.

---

> > > ### Author Response · Authors · 2023-11-21
> > > **Follow-up: more comparisons with existing methodologies**
> > >
> > > We thank the reviewer for their fast reply. Below, we make an effort to address the remaining concerns the reviewer had about the paper.
> > >
> > > **Presently, the author has introduced only a singular comparative method, a move that evidently lacks compelling strength.**
> > >
> > > We thank the reviewer for suggesting that additional comparisons are important for strengthening the paper. To make our empirical evaluation more convincing, we now extend our comparison to include several well-known tabular pretraining methods (including SubTab [1] which we already added): SCARF [2], VIME-self [3], DACL+ [4], which we selected as they are commonly compared against in the literature [1,22,23,24]. Furthermore, we also include comparisons against semi-supervised GBDTs (with pseudolabeling [21]), as suggested by Reviewer 1XYq.
> > >
> > > *Methodology*
> > > Following the methodology of Subsection 4.3.1, for SimCLR-Rank, SimSiam, SimCLR, SCARF, DACL+, VIME-self, SubTab, we pretrain for 300 epochs (note this is different from the setting of our previous responses). Note that SimCLR-Rank is 7x-14x faster than SimCLR/SCARF/DACL, as SCARF/DACL use the same loss as SimCLR (the augmentation strategy is different). For all methods, we finetune for 300 epochs following the methodology of Subsection 4.3.1.
> > >
> > > SimCLR-Rank, SimSiam, and SimCLR are hyperparameter tuned following the methodology in Subsection 4.3.1. VIME has the corruption probability tuned in [0.3, 0.5, 0.7]. DACL+ has $\alpha$ (the mixup amount) tuned in $[0.3, 0.6, 0.9]$ with $\rho$ (the binary masking probability) set to be $\alpha/2$. SCARF has its corruption probability tuned in $[0.3, 0.6, 0.9]$.
> > >
> > > Finally, we also include a semi-supervised GBDT baseline, as suggested by reviewer 1XYq, using pseudolabeling [21]. In pseudolabeling, we train a GBDT on the labeled dataset and then use this GBDT to produce labels for the unlabeled train set. Then we train the GBDT on this labeled + pseudolabeled train set.
> > >
> > > A GBDT ranker outputs real-valued scores. Given that pseudolabeling datasets using GBDTs in LTR is not a well-studied problem, we propose the following approach. To convert real-valued scores into relevance scores, we take the lowest output score when labeling the unlabeled train set, and subtract it from all real scores. Then we round these positive floats into their nearest integers, and let these be the pseudolabeled relevance scores. In MSLR, this results in the pseudolabels being integers from 0-7, in Istella 0-12. For Yahoo we increase the spread of the scores by multiplying the positive floats by 100 before rounding to the nearest integer (resulting in the pseudolabels being numbers from 0-5), since without doing this all the pseudolabels would be 0.
> > >
> > > We give the results in the following table (note that the GBDT baselines have stderr 0 due to being deterministic).
> > >
> > > | Method | MSLR | Yahoo | Istella |
> > > | - | - | - | - |
> > > | SimCLR-Rank| **0.3959 $\pm$ 0.0022** | 0.6022 $\pm$ 0.0013 |**0.5839 $\pm$ 0.0013** |
> > > | SimSiam    | 0.3935 $\pm$ 0.0034 | **0.6107 $\pm$ 0.0035** | 0.5618 $\pm$ 0.0049 |
> > > | SimCLR     | 0.3931 $\pm$ 0.0021 | 0.6015 $\pm$ 0.0023 | 0.5828 $\pm$ 0.0047 |
> > > | SCARF      | 0.3905 $\pm$ 0.0015 | 0.5976 $\pm$ 0.0028 | 0.5792 $\pm$ 0.0037 |
> > > | DACL+      | 0.3885 $\pm$ 0.0023 | 0.5770 $\pm$ 0.0049 | 0.5800 $\pm$ 0.0040 |
> > > | VIME-self   | 0.3757 $\pm$ 0.0127 | 0.5386 $\pm$ 0.0229 | 0.5361 $\pm$ 0.0080 |
> > > | SubTab      | 0.2879 $\pm$ 0.0019 | 0.5889 $\pm$ 0.0021 | 0.4700 $\pm$ 0.0031|
> > > | GBDT        | 0.3908 $\pm$ 0.0000 | 0.5932 $\pm$ 0.0000 | 0.5807 $\pm$ 0.0000 |
> > > | GBDT+pseudo | 0.3773 $\pm$ 0.0000 | 0.5967 $\pm$ 0.0000 | 0.5396 $\pm$ 0.0000 |
> > >
> > > Out of the pretraining methods we add to the comparisons, SCARF performs the best, though it still does not outperform SimCLR-Rank. Also recall that SimCLR-Rank is also 7-14x faster than SimCLR/SCARF/DACL+. We also find that pseudolabeling generally decreases GBDT performance. We believe this does not change the conclusions made in the paper. We hope these additional comparisons give the reviewer more confidence in our results.

---

> ### Author Response · Authors · 2023-11-21
> **Follow up: the impact of this work**
>
> **It becomes challenging to attribute a substantial level of innovation to this work.**
>
> We thank the reviewer for an opportunity for us to clarify the innovation in our work. SimCLR [14,15,16,17] is known to need large batch sizes to work well, which combined with its quadratic compute scaling in batch size makes it too inefficient. For example, pretraining SimCLR on CIFAR100/ImageNet requires many nodes of V100 GPUs [15].
>
> The naive way to make SimCLR more efficient is to sample the negatives, which prevents quadratic scaling in batch size. This is SimCLR-sample, which we already showed does not perform very well in MSLR/Istella (the two harder LTR public datasets). A more sophisticated approach is to only include hard negatives in the contrastive loss, which Robinson et al. [18] find makes contrastive learning much faster and also perform better.
>
> Our main technical insight (SimCLR-Rank) is that the hard negatives in LTR are already known–we do not have to mine for them at all because they are just samples in the same query group. Effectively, we utilize the structure of the LTR problem to obtain hard negative examples for free. Previous works showing how to mine hard negatives have been considered worthy contributions to the ML literature [18,19,20]. Note that the simplicity of SimCLR-Rank makes it more adaptable and thus more impactful: one can for example combine SCARF/DACL with SimCLR-Rank.
>
> We hope this response explains our position better on the technical contribution of our paper.
>
> **Confining the scope of this paper solely to its application in the niche domain of learning-to-rank significantly diminishes its impact.**
>
> We thank the reviewer for the opportunity to elaborate on the state of LTR in research and industry. LTR is a central component of highly important applications like search and recommendation, with large impacts on the economy and society. Many works solely on LTR have been published at top ML conferences [5,6,7,8,9,10,11,12, 28, 29, 30, 31]. There was also recently a workshop on ranking at ICML 2023 [13] (https://icml.cc/virtual/2023/workshop/21495). Burges et al. [11] won the ICML 2015 Test of Time award for their publication on LTR. So we believe significant results in the space of LTR are likely to achieve significant impact (and have done so in the past). LTR is a significant area in both machine learning research and industry.
>
> Recently, LTR has also garnered significant attention due to the use of new LLM (large language model) applications like Retrieval Augmented Generation (RAG) [25]. In RAG, a document retrieval mechanism is paired with an LLM to enhance the factual accuracy of an LLM’s response to a query. Recently, the document retrieval mechanism has included re-ranking [26] and tabular features [27]. As a result, we believe LTR is not only a significant area now but also will become even more important in the future.
>
> We appreciate the reviewer’s comments, as they have been highly helpful in improving our paper. If the concerns have been addressed, we would appreciate it if the reviewer could consider raising their score.

---

> > ### Comment · Reviewer_oW1i · 2023-11-21
> >
> > I must emphasize that, in my opinion, the substantial limitation of this paper arises from its exclusive focus on Learning to Rank (LTR) for tabular data. While the author provides numerous references related to LTR, it remains unclear how many of these references specifically pertain to tabular data exclusively. Perhaps my understanding of LTR has some deviation. Does LTR exclusively refer to Learning to Rank for tabular data?

---

> ### Author Response · Authors · 2023-11-21
> **Follow up: references**
>
> [1] SubTab: Subsetting Features of Tabular Data for Self-Supervised Representation Learning. Talip Ucar, Ehsan Hajiramezanali, Lindsay Edwards. NeurIPS 2019.
> [2] SCARF: Self-Supervised Contrastive Learning using Random Feature Corruption. Dara Bahri, Heinrich Jiang, Yi Tay, Donald Metzler. ICLR 2022.
> [3] VIME: Extending the Success of Self- and Semi-supervised Learning to Tabular Domain. Jinsung Yoon, Yao Zhang, James Jordon, Mihaela van der Schaar. NeurIPS 2020.
> [4] Towards Domain-Agnostic Contrastive Learning. Vikas Verma, Minh-Thang Luong, Kenji Kawaguchi, Hieu Pham, Quoc V. Le. ICML 2021.
> [5] Are Neural Rankers still Outperformed by Gradient Boosted Decision Trees?, Qin et al., ICLR 2021.
> [6] Toward Understanding Privileged Features Distillation in Learning-to-Rank., Yang et al., NeurIPS 2022.
> [7] PiRank: Scalable Learning To Rank via Differentiable Sorting. Robin Swezey, Aditya Grover, Bruno Charron, Stefano Ermon. NeurIPS 2021.
> [8] On the Value of Prior in Online Learning to Rank. Branislav Kveton, Ofer Meshi, Masrour Zoghi, Zhen Qin. AISTATS 2022.
> [9] Learning to rank using gradient descent. Learning to rank using gradient descent. Chris Burges, Tal Shaked, Erin Renshaw, Ari Lazier, Matt Deeds, Nicole Hamilton, Greg Hullender. ICML 2005.
> [10] Learning to rank: from pairwise approach to listwise approach. Zhe Cao, Tao Qin, Tie-Yan Liu, Ming-Feng Tsai, Hang Li. ICML 2007.
> [11] Listwise approach to learning to rank: theory and algorithm. Fen Xia, Tie-Yan Liu, Jue Wang, Wensheng Zhang, Hang Li. ICML 2008.
> [12] Ranking measures and loss functions in learning to rank. Wei Chen, Tie-Yan Liu, Yanyan Lan, Zhi-ming Ma, Hang Li. NeurIPS 2009.
> [13] The Many Facets of Preference-Based Learning. Aadirupa Saha, Mohammad Ghavamzadeh, Robert Busa-Fekete, Branislav Kveton, Viktor Bengs. ICML 2023.
> [14] A Simple Framework for Contrastive Learning of Visual Representations. Ting Chen, Simon Kornblith, Mohammad Norouzi, Geoffrey Hinton. ICML 2020.
> [15] Exploring Simple Siamese Representation Learning, invited talk https://rosanneliu.com/dlctfs/dlct_210326.pdf.
> [16] Exploring Simple Siamese Representation Learning. Xinlei Chen, Kaiming He. CVPR 2021.
> [17] Bootstrap Your Own Latent A New Approach to Self-Supervised Learning. Grill et al. NeurIPS 2020.
> [18] Contrastive learning with hard negative samples. Joshua Robinson, Ching-Yao Chuang, Suvrit Sra, Stefanie Jegelka. ICLR 2021.
> [19] Debiased Contrastive Learning. Ching-Yao Chuang, Joshua Robinson, Lin Yen-Chen, Antonio Torralba, Stefanie Jegelka. NeurIPS 2020.
> [20] Robust Contrastive Learning Using Negative Samples with Diminished Semantics. Songwei Ge, Shlok Mishra, Chun-Liang Li, Haohan Wang, David Jacobs. NeurIPS 2021.
> [21] Self-training with Noisy Student improves ImageNet classification. Qizhe Xie, Minh-Thang Luong, Eduard Hovy, Quoc V. Le. CVPR 2020.
> [22] STab: Self-supervised Learning for Tabular Data. Ehsan Hajiramezanali, Nathaniel Diamant, Gabriele Scalia, Max W. Shen.
> [23] Transfer Learning with Deep Tabular Models. Levin et al., ICLR 2023.
> [24] MET: Masked Encoding for Tabular Data. Kushal Majmundar, Sachin Goyal, Praneeth Netrapalli, Prateek Jain.
> [25] Retrieval-Augmented Generation for Knowledge-Intensive NLP Tasks.
> [26] Re2G: Retrieve, Rerank, Generate. Michael Glass, Gaetano Rossiello, Md Faisal Mahbub Chowdhury, Ankita Rajaram Naik, Pengshan Cai, Alfio Gliozzo.
> [27] T-RAG: End-to-End Table Question Answering via Retrieval-Augmented Generation. Feifei Pan, Mustafa Canim, Michael Glass, Alfio Gliozzo, James Hendler.
> [28] Beyond Greedy Ranking: Slate Optimization via List-CVAE. Ray Jiang, Sven Gowal, Timothy A. Mann, Danilo J. Rezende. ICLR 2019.
> [29] Towards Amortized Ranking-Critical Training for Collaborative Filtering. Sam Lobel, Chunyuan Li, Jianfeng Gao, Lawrence Carin. ICLR 2020.
> [30] Individually Fair Ranking. Amanda Bower, Hamid Eftekhari, Mikhail Yurochkin, Yuekai Sun. ICLR 2021.
> [31] Adversarial Retriever-Ranker for dense text retrieval. Hang Zhang, Yeyun Gong, Yelong Shen, Jiancheng Lv, Nan Duan, Weizhu Chen. ICLR 2022.

---

> ### Author Response · Authors · 2023-11-21
> **Follow up on the topic of tabular LTR**
>
> We thank the reviewer for their time and consideration.
>
> LTR is a general problem and has been applied to tabular, text, images and various other domains. Among the papers we cited these focus exclusively on tabular LTR: [4,5,6,7,8,9,10,11,12] (all papers were published at ICLR, NeurIPS, ICML). One of these is Burges et al. [9], which won the ICML 2015 Test of Time Award. To give some more context, a significant portion of research and real-world applications of LTR have focused on tabular LTR due to its efficiency at web-scale and ubiquitous nature of tabular data.
>
> We would also like to note that many impactful papers on pretraining published at top ML conferences also focused on only one modality and application: images [14,16,17,20,21], text [13,15,18,19], tabular [1,2,3]. Hence we strongly believe that studying the very practical (as illustrated in experiments) topic of pretraining for tabular LTR can also be similarly impactful and useful to the community.
>
> We thank the reviewer for their continued engagement throughout the discussion period, and we hope that we have addressed the reviewer’s concerns.
>
>
>
> [1] SubTab: Subsetting Features of Tabular Data for Self-Supervised Representation Learning. Talip Ucar, Ehsan Hajiramezanali, Lindsay Edwards. NeurIPS 2019.
> [2] SCARF: Self-Supervised Contrastive Learning using Random Feature Corruption. Dara Bahri, Heinrich Jiang, Yi Tay, Donald Metzler. ICLR 2022.
> [3] VIME: Extending the Success of Self- and Semi-supervised Learning to Tabular Domain. Jinsung Yoon, Yao Zhang, James Jordon, Mihaela van der Schaar. NeurIPS 2020.
> [4] Individually Fair Ranking. Amanda Bower, Hamid Eftekhari, Mikhail Yurochkin, Yuekai Sun. ICLR 2021.
> [5] Are Neural Rankers still Outperformed by Gradient Boosted Decision Trees?, Qin et al., ICLR 2021.
> [6] Toward Understanding Privileged Features Distillation in Learning-to-Rank., Yang et al., NeurIPS 2022.
> [7] PiRank: Scalable Learning To Rank via Differentiable Sorting. Robin Swezey, Aditya Grover, Bruno Charron, Stefano Ermon. NeurIPS 2021.
> [8] On the Value of Prior in Online Learning to Rank. Branislav Kveton, Ofer Meshi, Masrour Zoghi, Zhen Qin. AISTATS 2022.
> [9] Learning to rank using gradient descent. Chris Burges, Tal Shaked, Erin Renshaw, Ari Lazier, Matt Deeds, Nicole Hamilton, Greg Hullender. ICML 2005.
> [10] Learning to rank: from pairwise approach to listwise approach. Zhe Cao, Tao Qin, Tie-Yan Liu, Ming-Feng Tsai, Hang Li. ICML 2007.
> [11] Listwise approach to learning to rank: theory and algorithm. Fen Xia, Tie-Yan Liu, Jue Wang, Wensheng Zhang, Hang Li. ICML 2008.
> [12] Ranking measures and loss functions in learning to rank. Wei Chen, Tie-Yan Liu, Yanyan Lan, Zhi-ming Ma, Hang Li. NeurIPS 2009.
> [13] Universal Language Model Fine-tuning for Text Classification. Jeremy Howard, Sebastian Ruder. ACL 2018.
> [14] A Simple Framework for Contrastive Learning of Visual Representations. Ting Chen, Simon Kornblith, Mohammad Norouzi, Geoffrey Hinton. ICML 2020.
> [15] Multilingual Constituency Parsing with Self-Attention and Pre-Training. Nikita Kitaev, Steven Cao, Dan Klein. ACL 2019.
> [16] Exploring Simple Siamese Representation Learning. Xinlei Chen, Kaiming He. CVPR 2021.
> [17] Bootstrap Your Own Latent A New Approach to Self-Supervised Learning. Grill et al. NeurIPS 2020.
> [18] Adversarial Multi-task Learning for Text Classification. Pengfei Liu, Xipeng Qiu, Xuanjing Huang. ACL 2017.
> [19] Unsupervised Sparse Vector Densification for Short Text Similarity. Yangqiu Song, Dan Roth. NAACL 2015.
> [20] Robust Contrastive Learning Using Negative Samples with Diminished Semantics. Songwei Ge, Shlok Mishra, Chun-Liang Li, Haohan Wang, David Jacobs. NeurIPS 2021.
> [21] Self-training with Noisy Student improves ImageNet classification. Qizhe Xie, Minh-Thang Luong, Eduard Hovy, Quoc V. Le. CVPR 2020.

---

> > ### Comment · Reviewer_oW1i · 2023-11-22
> >
> > Thank you for your detailed response. I have one more question: There are currently some studies on pretraining neural networks on tabular data, which also demonstrate improved performance. Do existing tabular data pretraining methods also involve the application of methods like SimCLR or SimSiam? From my understanding, these two methods in your paper do not consider the nature of LTR and are similar to general tabular data.

---

> > > ### Author Response · Authors · 2023-11-22
> > >
> > > We thank the reviewer for the insightful question and comments.
> > >
> > > **Do existing tabular data pretraining methods also involve the application of methods like SimCLR or SimSiam?**
> > >
> > > Some well-known tabular data pretraining methods do not use SimCLR [5] or SimSiam [6], notably SubTab (NeurIPS 2021) [1] and VIME (NeurIPS 2020) [3]. Other well-known tabular pretraining methods, DACL+ (ICML 2021) [4] and SCARF (ICLR 2022 spotlight) [2] are essentially SimCLR combined with tabular-specific augmentation methods. Note that in our comprehensive comparison https://openreview.net/forum?id=Dk1ybhMrJv&noteId=1Z4qMYixVv we find that the simple augmentations considered in our paper (zeros and gaussian noise) actually worked better than the more complicated augmentations of DACL+ and SCARF. DACL+/SCARF/VIME/SubTab are the methods most commonly referenced and compared against in the literature, as we noted in our earlier replies.
> > >
> > > We would also like to finally bring to the reviewer’s attention that LTR is very different from the settings considered in SubTab/DACL+/SCARF/VIME papers. They consider tabular classification datasets, while LTR labels have an ordering (i.e. 4 > 3 > 2 > 1 > 0), which makes it somewhat like a regression problem. Methods that work in classification may not generalize to other problems.
> > >
> > > **From my understanding, these two methods in your paper (SimCLR, SimSiam) do not consider the nature of LTR and are similar to general tabular data.**
> > >
> > > As the reviewer correctly observed, SimCLR and SimSiam are methods that were not specifically designed to consider the nature of tabular LTR or tabular data. However, tabular data is very general: any fixed dimensional data can be considered tabular data. For example, images can be configured into tables by arranging pixel values into columns (see the DACL+ and SubTab papers).
> > >
> > > Because tabular data is so diverse and general (it could be anything fixed-dimensional!) tabular pretraining methods are also very general. It is unclear whether SubTab/DACL+/SCARF/VIME (the well-known tabular pretraining methods) or SimCLR/SimSiam are better designed for tabular data. All of these methods take in fixed-dimensional inputs and can be applied to a wide variety of data.
> > >
> > > Ultimately, in the tabular LTR setting, our extensive experimental comparisons ( https://openreview.net/forum?id=Dk1ybhMrJv&noteId=1Z4qMYixVv) show that SimCLR and SimSiam do often perform better than previous work on tabular pretraining. Furthermore, SimCLR-Rank (which does consider LTR structure) along with SimSiam perform the best.
> > >
> > > We hope these responses were helpful. Please let us know if you have any further questions.
> > >
> > > [1] SubTab: Subsetting Features of Tabular Data for Self-Supervised Representation Learning. Talip Ucar, Ehsan Hajiramezanali, Lindsay Edwards. NeurIPS 2021.
> > > [2] SCARF: Self-Supervised Contrastive Learning using Random Feature Corruption. Dara Bahri, Heinrich Jiang, Yi Tay, Donald Metzler. ICLR 2022.
> > > [3] VIME: Extending the Success of Self- and Semi-supervised Learning to Tabular Domain. Jinsung Yoon, Yao Zhang, James Jordon, Mihaela van der Schaar. NeurIPS 2020.
> > > [4] Towards Domain-Agnostic Contrastive Learning. Vikas Verma, Minh-Thang Luong, Kenji Kawaguchi, Hieu Pham, Quoc V. Le. ICML 2021.
> > > [5] A Simple Framework for Contrastive Learning of Visual Representations. Ting Chen, Simon Kornblith, Mohammad Norouzi, Geoffrey Hinton. ICML 2020.
> > > [6] Exploring Simple Siamese Representation Learning. Xinlei Chen, Kaiming He. CVPR 2021.

---

> > > > ### Author Response · Authors · 2023-11-22
> > > > **Gentle request on final feedback**
> > > >
> > > > Thank you for your time and consistent effort throughout this discussion! Your thoughtful comments have been very helpful in improving our paper. Today is the end of the discussion period, and we are eagerly waiting to hear any additional feedback you may still have about the paper.
> > > >
> > > > We hope that we have answered your last question on SimCLR/SimSiam. Please let us know if there’s anything more we can do to improve our paper and/or increase our score.

---

> > > > > ### Comment · Reviewer_oW1i · 2023-11-23
> > > > >
> > > > > I appreciate the author's response. Given that SimCLR has been validated to exhibit certain advantages on general tabular data, it is not surprising that it brings about a performance enhancement in tabular data Learning to Rank (LTR). I find that the revelations in both SimCLR and SimSiam do not offer many novel insights. On the methodological front, the author acknowledges that SimCLR-Rank's performance is not significantly stronger, and the method bears a striking resemblance to SimCLR. In summary, this paper conducts extensive experimental analysis in neural network pretraining on tabular data, potentially yielding positive impacts on the field. I maintain the borderline reject rating, but accepting the paper would not pose significant issues in my view.

---

> ### Author Response · Authors · 2023-11-23
>
> Thank you for your feedback. We would like to clear up one last bit of confusion.
>
> SimCLR-Rank is 7-14x faster wall-clock time and has an order-wise (big-O) speed/memory improvement over SimCLR (see Subsection 3.1). This is a significant improvement over SimCLR, which is known to require many nodes of V100 GPUs to train datasets like ImageNet (check the SimSiam talk slides https://rosanneliu.com/dlctfs/dlct_210326.pdf).
>
> **SimCLR is simply not practical in the big data regime (like large-scale recommendations/search/LTR), while SimCLR-Rank is, due to the speed difference**. We hope the reviewer can incorporate this into their final evaluation.

---

### Official Review · Reviewer_1XYq · 2023-11-06

**Soundness:** 2 fair
**Presentation:** 2 fair
**Contribution:** 2 fair
**Rating:** 5
**Confidence:** 3

**Summary:**

This paper studies the effectiveness of unsupervised pretraining in tabular Learning-To-Rank (LTR) problems. First, authors show how two well-known self-supervised training methods (SimCLR, SimSiam) can be formulated in LTR problems. Next, authors proposes SimCLR-Rank, which is a variant of SimCLR. SimCLR-Rank sample negatives in SimCLR loss only from the same query group. Experimentally, authors show that pretrained models can outperform GBDTs, which is one of the strongest baselines, under label scarcity setting. Additionally, authors compare self-supervised methods (SimCLR, SimSiam, SimCLR-Rank) in various settings (datasets, data augmentation methods, outlier NDCG, etc.)

**Strengths:**

- Representation learning in tabular LTR has not been studied much.
- SimCLR-Rank provides a strategy specific to the structure of LTR task.

**Weaknesses:**

- Technical novelty is limited.
- Datasets in experiments are limited to three datasets and one private dataset.
- Representation learning on tabular data not necessarily needs to consider LTR setting, since multi-layer MLP will be finetuned for LTR task. As a paper that proposes a new tabular self-supervised learning method, it lacks the comparison with other existing methods, such as
    - Hajiramezanali, E., Diamant, N. L., Scalia, G., & Shen, M. W. (2022, October). STab: Self-supervised Learning for Tabular Data. In *NeurIPS 2022 First Table Representation Workshop*.
    - Wang, W., KIM, B. H., & Ganapathi, V. (2022). RegCLR: A Self-Supervised Framework for Tabular Representation Learning in the Wild.
    - Ucar, T., Hajiramezanali, E., & Edwards, L. (2021). Subtab: Subsetting features of tabular data for self-supervised representation learning. *Advances in Neural Information Processing Systems*

**Questions:**

- What’s the intuition behind SimCLR-Rank working better?
- Why does pretraining improve outlier NDCG?
- Typically, pretraining has advantages by learning useful representations from large datasets, while this paper seems to use unlabeled data in the same downstream task for pretraining. Am I understanding correctly? Can pretraining in one downstream dataset transfer to other datasets?
- GBDT could use some well-known semi-supervised learning methods such as pseudo-labeling or consistency regularization to use unlabeled dataset. Do the pretraining methods still outperform GBDT with semi-supervised learning?

---

> ### Author Response · Authors · 2023-11-19
> **Reply part 1**
>
> Thank you for your review and feedback. We would like to first take an opportunity to clarify the impact and contributions of our work.
>
> We strongly believe our work will significantly impact industries that work with LTR problems (search, recommendation), by causing them to revise, or at least revisit, current SOTA methods. Today many large companies use GBDTs for ranking tabular data, and this approach is SOTA even in the research community. The novel starting point of our work is the often ignored fact that most of the data collected in LTR systems is unlabeled. We have shown through extensive experiments on standard public datasets and a large-scale private dataset that pretrained deep rankers can leverage unlabeled data to consistently outperform GBDTs, with outsized gains in robustness towards outlier data.
>
> All reviewers agree we have convincingly demonstrated this. Reviewer 1XYq: “Experimentally, authors show that pretrained models can outperform GBDTs, which is one of the strongest baselines, under label scarcity setting.” Reviewer oW1i: “the paper highlights the superior performance of pretrained deep rankers, especially on outlier queries, in scenarios with limited labeled data.” Reviewer i3NM: “The major contribution of the paper is to identify areas where [pretrained deep rankers] help, including query sparsity and label sparisity scenarios.”
>
> We now address the listed concerns one by one.
>
> **Technical novelty is limited.**
>
> We thank the reviewer for their feedback.  We would like to bring to the reviewer’s attention these following novel contributions:
>
> 1. We make the observation that in Learning-To-Rank (LTR) applications like search and recommendations, there is an abundance of unlabeled data. Before our work, this LTR problem setting has not been studied, to the best of our knowledge. **Reviewers agree on this point.** Reviewer 1XYq: “Representation learning in tabular LTR has not been studied much.” Reviewer oW1i: “The paper studies pre-trained DNN for tabular LTR problems, which is an underexplored problem.” Reviewer i3NM: “To the reviewer’s knowledge, this is the first work that shows some promises for pre-trained DNNs for the LTR task.”
>
> 2. SimCLR-Rank is the first pretraining method that specifically leverages LTR problem structure.  SimCLR, as noted by reviewer i3NM, is very slow: “the major weakness of SimCLR for non-ranking problems was complexity.”  Our method SimCLR-Rank leverages the unique structure of the LTR problem to gain orders of magnitude speed improvements (Table 6 on page 16 of the paper) while performing slightly better than SimCLR (Table 5 on page 9 of the paper).  We believe this is technically novel.
>
> 3. We would finally like to emphasize that our results on the large-scale industry-size dataset are a big contribution. Most LTR papers [1,2,3] do not give results on industry-scale datasets, and do not prove their hypotheses on true large scale datasets because of lack of availability of such data and prohibitively large expense of running large-scale experiments. The fact that we demonstrate pretraining rankers achieve SOTA on public datasets in LTR but also **emphatically** on a large-scale commercial dataset is highly impactful. Therefore we believe our results will help the LTR and the wider Tabular-DL communities tremendously.
>
> **Datasets in experiments are limited to three datasets and one private dataset.**
>
> We thank the reviewer for giving us an opportunity to clarify the experimental breadth. In LTR literature experiments these three datasets are generally considered the standard, with many influential and peer-reviewed work only evaluating at most the three public datasets we experiment on [1, 2, 3, 4]. Further, tabular (a.k.a. numerical) ML (e.g. LTR) systems are important for modern enterprises and therefore validating research hypotheses on large-scale practical datasets is valuable. Unfortunately, this is challenging due to the expensive and proprietary nature of such data. Given that we evaluate on all three of the standard public datasets—plus a large private dataset—we believe our empirical evaluation is stronger than comparable papers in the literature and could be valuable for the vast LTR and Tabular-DL communities.

---

> ### Author Response · Authors · 2023-11-19
> **Reply part 2**
>
> **Lacking comparison and discussion with existing tabular SSL methods.**
>
> We thank the reviewer for bringing up the comparison with existing tabular SSL methods.  We add a comprehensive and expanded related works section to section A.1 (in the appendix of the paper, colored in blue) discussing the current state of tabular SSL.  After the paper decision, we plan to reorder the paper to place it into the main text.
>
> We would like to clarify that our primary goal is to demonstrate that DL rankers built using pre-training techniques can achieve SOTA performance in LTR settings, and consequently beat GBDT models (prior SOTA). Only as a secondary goal, we investigate a few simple but effective pre-training strategies such as SimSiam, SimCLR, and our variant SimCLR-Rank. While we identified SimCLR-Rank has a positive impact in a wide variety of LTR scenarios, we also show that SimSiam works better for our large-scale proprietary dataset.  Our goal is not to show that any one particular method is the best for pretraining in LTR.
>
> Regarding the suggested tabular SSL methods: STab [12] applies a SimSiam-like method to tabular self-supervised learning, RegCLR [13] aims to extract tables from images and is not applicable to our setting, and SubTab [14] is a well-known and influential tabular SSL technique.  Given that STab has a similar strategy as SimSiam which we already evaluate in our paper (and because there was no published code for STab), we focus on evaluating SubTab in this response (respecting the time constraints of the rebuttal).
>
> We evaluate SubTab [14] in the setting of Subsection 4.3.2.  SubTab’s pretraining objective divides input features into multiple subsets (in the language of computer vision, “multiple views”) and trains an autoencoder to reconstruct the original input features.  To pretrain using SubTab, we divide the input features into 4 subsets with 75% overlap, and with input corruptions of 15% masking probability and Gaussian noise of scale 0.1 (suggested in the SubTab paper as a good setting).  The finetuning strategy and the choice for encoder model is the same as for SimCLR-Rank.  We find that SubTab performs respectably on Yahoo, but fails to learn the MSLR and Istella datasets.
>
> | Method | MSLR | Yahoo | Istella |
> | --- | ----------- | --- | ---- |
> | SimCLR-Rank | **0.3929 $\pm$ 0.0018** | **0.5989 $\pm$ 0.0010** | **0.5830 $\pm$ 0.0013** |
> | SubTab | 0.2879 $\pm$ 0.0019 | 0.5889 $\pm$ 0.0021 | 0.4700 $\pm$ 0.0031 |
>
> **What’s the intuition behind SimCLR-Rank working better?**
>
> Thank you for giving us the opportunity to clarify. Recalling the response to the previous concern, our goal is to show that pretraining helps in LTR, not that any particular pretraining method (including SimCLR-Rank) is better–although SimCLR-Rank does perform well in many different settings in our experiments.
>
> SimCLR-Rank is different from SimCLR in two aspects: (1) it uses items from the same query group as freely available hard-negatives, (2) it reduces the number of negatives from batchsize to at most query group size and thus allow better computational scaling with batchsize. To identify which of these contribute more towards SimCLR-Rank’s performance, we performed an additional experiment below (as suggested by reviewer i3NM), comparing SimCLR-Rank with a variant of SimCLR (which we call as SimCLR-sample). In SimCLR-sample, instead of using all the other items as negatives, we uniformly sample a constant number of negatives from the batch. This reduces the computational complexity from quadratic in batchsize (of SimCLR) to linear in batchsize. In the below experiments we follow the setup in Subsection 4.3.2 and SimCLR-Rank and SimCLR-sample use the same number of negatives and batchsize.
>
> We find that SimCLR-Rank performs better on MSLR/Istella, while SimCLR-sample performs better on Yahoo. We note that MSLR/Istella are much sparser than Yahoo (see Table 12 in the paper), and are more representative of typical search and recommendations applications. Therefore we still believe that SimCLR-Rank which selects hard-negatives from the same query group is a useful pre-training method for the LTR toolbox and can be highly effective in many situations.
>
> | Method | MSLR | Yahoo | Istella |
> | --- | ----------- | --- | ---- |
> | SimCLR-Rank | **0.3929 $\pm$ 0.0018** | 0.5989 $\pm$ 0.0010 | **0.5830 $\pm$ 0.0013** |
> | SimCLR-sample | 0.3890 $\pm$ 0.0008 | **0.6056 $\pm$ 0.0047** | 0.5787 $\pm$ 0.00351 |

---

> ### Author Response · Authors · 2023-11-19
> **Reply part 3**
>
> **Why does pretraining improve outlier NDCG?**
>
> It is a well-known phenomenon that pretraining helps improve robustness and out of distribution performance in domains like image and text [5, 6]. The same phenomenon, we find, is also true in the tabular LTR setting. Liu et al., [7] find that pretraining with data augmentation allows models to learn richer features they can generalize to outlier or out of distribution examples better–merely supervised training only incentivizes models to attend to highly predictive features in the training set.  We conjecture that pretraining with augmentations in LTR (as we do in our paper) improves performance on outlier NDCG for similar reasons.
>
> **Can pretraining in one downstream dataset transfer to other datasets?**
>
> We thank the reviewer for bringing up this point. Tabular datasets have specific features which can be very different. In fact, each tabular dataset is its own domain. Because in general there are no common features between tabular datasets, it is not clear how to have common knowledge. Levin et al. [8] show that if features are shared, it is possible to transfer knowledge between datasets, but in general it is not known how to do transfer learning in tabular data.
>
> **Do the pretraining methods still outperform GBDT with semi-supervised learning?**
>
> We thank the reviewer for bringing up important baselines. Rubachev et al. [9] find that consistency regularization does not help GBDTs (https://openreview.net/forum?id=kjPLodRa0n&noteId=E4FT6VCltG). As suggested by the reviewer, we ran new experiments on pseudolabeling for GBDTs, and found that it (typically) performs worse than the GBDT baselines themselves.
>
> Methodology: we allow 0.2% of the query groups to be labeled in each dataset, we train a GBDT on the labeled set and then use the GBDT to label the unlabeled train set.
>
> A GBDT ranker outputs real-valued scores. Given that pseudolabeling datasets using GBDTs in LTR is not a well-studied problem, we propose the following approach. To convert real-valued scores into relevance scores, we take the lowest output score when labeling the unlabeled train set, and subtract it from all real scores. Then we round these positive floats into their nearest integers, and let these be the pseudolabeled relevance scores. In MSLR, this results in the pseudolabels being integers from 0-7, in Istella 0-12. For Yahoo we increase the spread of the scores by multiplying the positive floats by 100 before rounding to the nearest integer (resulting in the pseudolabels being numbers from 0-5), since without doing this all the pseudolabels would be 0.
>
> We find that overall, pseudolabeling decreases the performance of GBDTs on MSLR and Istella (the sparser datasets which are typically more representative of search and recommendations applications, check Table 10 in the paper) while slightly improving the performance for Yahoo, which is somewhat an “easy” dataset. All the GBDT results below underperform pretrained deep models (we use the methodology in Section 4.1.1).  Note that due to the size of the dataset, the GBDT results are deterministic (no stderr).
>
> | Method |MSLR | Yahoo | Istella |
> | --- | ----------- | - | - |
> | GBDT |0.3908|0.5932|0.5807|
> | GBDT + pseudo-labeling| 0.3773 | 0.5967 | 0.5396|
> | Pretrained DL | **0.3959 $\pm$ 0.0022** | **0.6107 $\pm$ 0.0035** | **0.5839 $\pm$ 0.0049** |
>
> We thank the reviewer for their consideration, and are happy to help with any further concerns. If we have addressed the concerns we would appreciate it if the reviewer could raise their score.

---

> ### Author Response · Authors · 2023-11-19
> **Reply part 4**
>
> [1] Are Neural Rankers still Outperformed by Gradient Boosted Decision Trees?, Qin et al., ICLR 2021.
> [2] Learning Groupwise Multivariate Scoring Functions Using Deep Neural Networks., Ai et al., SIGIR 2019.
> [3] SetRank: Learning a Permutation-Invariant Ranking Model for Information Retrieval., Pang et al., SIGIR 2020.
> [4] Toward Understanding Privileged Features Distillation in Learning-to-Rank., Yang et al., NeurIPS 2022
> [5] Using Self-Supervised Learning Can Improve Model Robustness and Uncertainty., Dan Hendrycks, Mantas Mazeika, Saurav Kadavath, Dawn Song. NeurIPS 2019.
> [6] Pretrained Transformers Improve Out-of-Distribution Robustness., Dan Hendrycks, Xiaoyuan Liu, Eric Wallace, Adam Dziedzic, Rishabh Krishnan, Dawn Song. ACL 2020.
> [7] Self-supervised Learning is More Robust to Dataset Imbalance., Hong Liu, Jeff Z. HaoChen, Adrien Gaidon, Tengyu Ma. ICLR 2022.
> [8] Transfer Learning with Deep Tabular Models. Roman Levin, Valeriia Cherepanova, Avi Schwarzschild, Arpit Bansal, C. Bayan Bruss, Tom Goldstein, Andrew Gordon Wilson, Micah Goldblum. ICLR 2023.
> [9] Revisiting Pretraining Objectives for Tabular Deep Learning. Ivan Rubachev, Artem Alekberov, Yury Gorishniy, Artem Babenko.
> [10] A Simple Framework for Contrastive Learning of Visual Representations. Ting Chen, Simon Kornblith, Mohammad Norouzi, Geoffrey Hinton. ICML 2020.
> [11] Exploring Simple Siamese Representation Learning. Xinlei Chen, Kaiming He. CVPR 2021.
> [12] STab: Self-supervised Learning for Tabular Data. Hajiramezanali, E., Diamant, N. L., Scalia, G., & Shen, M. W. NeurIPS 2022 First Table Representation Workshop.
> [13] RegCLR: A Self-Supervised Framework for Tabular Representation Learning in the Wild.Wang, W., KIM, B. H., & Ganapathi.
> [14] Subtab: Subsetting features of tabular data for self-supervised representation learning. Ucar, T., Hajiramezanali, E., & Edwards, L.. NeurIPS 2021.
> [15] Improving Out-of-Distribution Robustness via Selective Augmentation.  Huaxiu Yao, Yu Wang, Sai Li, Linjun Zhang, Weixin Liang, James Zou, Chelsea Finn. ICML 2022

---

> > ### Comment · Reviewer_1XYq · 2023-11-19
> >
> > Thank you for providing the detailed response and updated results. I appreciate the authors’ efforts in addressing some of my questions.
> >
> > I have an additional question on the semi-supervised experiment result: 0.2% of the query groups look pretty small though I understand this setting is used in other experiments as well. What does the curve look like, when varying the ratio of labeled query groups?

---

> > > ### Author Response · Authors · 2023-11-19
> > > **Re: semi-supervised GBDTs with larger % of labeled query groups**
> > >
> > > We thank the reviewer for their prompt reply.  Below, we provide the change on GBDT NDCG after applying pseudolabeling with different percentages of labeled query groups:
> > >
> > > | Dataset    | Label fraction=0.2% |Label fraction=1% | Label fraction=10% | Label fraction=50% | Label fraction=80% |
> > > | ----------- | -| ----------- | - | - | - |
> > > | MSLR     | -0.0135 | -0.0375       | -0.0177 | -0.0673 | -0.0577|
> > > | Yahoo   | +0.0035| -0.0439       | -0.0272 | -0.0007 | -0.0028 |
> > > | Istella | -0.0411 |-0.0678 | -0.0017 | -0.005 | -0.0033 |
> > >
> > > After evaluation, we observe: (1) pseudolabeling generally does not help, and often hurts GBDT performance, (2) the negative effect tends to decrease as we increase the label fraction (because the amount of pseudolabeling decreases), and (3) pseudolabeling performs poorly on MSLR (which is the hardest dataset of the three).  We thank the reviewer for recommending this important baseline, which is a helpful contribution towards strengthening the arguments in our paper.

---

> > > > ### Comment · Reviewer_1XYq · 2023-11-19
> > > >
> > > > Thank you for your prompt update! I have adjusted my score to 5 as the authors have adeptly addressed some of the concerns I raised. However, I still maintain two primary concerns with this paper.
> > > >
> > > > - While I acknowledge the significance of Learning to Rank (LTR) in real-world applications, the paper falls short in demonstrating substantial technical novelty. It's well-established that deep learning models often benefit from pretraining across various domains such as vision, NLP, video, graph, and time series. As a result, the technical contributions of this work seem somewhat limited in comparison to existing practices in other domains.
> > > > - Moreover, unlike many other domains where pretrained models exhibit transferability across datasets, the suggested method in this paper lacks such transferability. Transferability is a fundamental feature of pretrained models that has greatly contributed to recent advancements in machine learning. To make this work more impactful in the realm of LTR, I believe it is imperative for the authors to conduct further exploration on the transferability of their proposed method across different datasets by developing additional techniques.

---

> > > > > ### Author Response · Authors · 2023-11-22
> > > > > **Follow up**
> > > > >
> > > > > Thank you very much for engaging with us in the discussion, we very much appreciate it. We also understand and appreciate your viewpoint. We would like to take an opportunity to explain our contributions better.
> > > > >
> > > > > **It's well-established that deep learning models often benefit from pretraining across various domains such as vision, NLP, video, graph, and time series. As a result, the technical contributions of this work seem somewhat limited in comparison to existing practices in other domains.**
> > > > >
> > > > > *First*, we feel that algorithmic novelty is not necessary for a result to be novel. For instance, if one were to show that logistic regression outperforms SOTA deep models for an image task on a particular class of practically-relevant images, this would be a novel result, despite logistic regression already existing.
> > > > >
> > > > > Our work follows in this spirit. As the reviewer correctly states, many published works have shown that (unsupervised) pretraining can work in a variety of domains, across images [2,3,8,9], text [10,11], graphs [12], and time series [1,4,5]. But what works in other domains *might not* work for tabular LTR. For example, it is common practice to use features directly from pretrained encoders for downstream tasks in images [2,3,8,9], text [15], graphs [12], time series [4,5], and tabular data [7,16,17]. But we found that this common practice actually performs very poorly in LTR (Subsection 4.3.1).
> > > > >
> > > > > Without our extensive experimental exploration, it would still remain unclear how to enable deep rankers to outperform GBDTs in LTR. It has been an open question in the literature for at least the past 15 years whether deep learning could outperform GBDTs for tabular LTR [18, 28,29,30]. We demonstrate that simple techniques (some of them already existing, some modifications proposed by us) can solve this difficult problem. We find this to be very exciting, and something that will be of great interest to the machine learning and ranking communities.
> > > > >
> > > > > *Second*, we would like to clarify the technical contribution of SimCLR-Rank.
> > > > >
> > > > > SimCLR is known to need large batch sizes to work well [3,19], which combined with its quadratic compute scaling in batch size is too inefficient (in terms of GPU-hours) to apply in real-world applications. The naive way to make SimCLR more efficient is to sample the negatives, which prevents quadratic scaling in batch size (i.e. SimCLR-sample, which we showed does not work well). A more sophisticated approach is to only include hard negatives in the contrastive loss, which Robinson et al. [20] find makes contrastive learning faster and better.
> > > > >
> > > > > Our main technical insight (SimCLR-Rank) is to utilize the structure of the LTR problem to obtain hard negative examples for free. We show that SimCLR-Rank is able to outperform previous baselines (for more evidence, see our new extensive experimental comparison https://openreview.net/forum?id=Dk1ybhMrJv&noteId=1Z4qMYixVv). Previous works showing how to mine hard negatives have been considered worthy contributions to the ML literature [20,21,22].
> > > > >
> > > > > **To make this work more impactful in the realm of LTR, I believe it is imperative for the authors to conduct further exploration on the transferability of their proposed method across different datasets by developing additional techniques.**
> > > > >
> > > > > We thank the reviewer for bringing up this important point, and agree that transferability would be a highly valued contribution.
> > > > >
> > > > > We would like to point out that it is common for unsupervised pretraining (self-supervised learning) papers to not consider transferability (including in very high-impact and influential works). Examples include: tabular classification [7,16,17,23], image classification [20,21,22], and text classification [20,21]. Like these papers, we show that pretraining in the same dataset provides significant benefits, and transfer learning is not needed to achieve these results.
> > > > >
> > > > > In tabular data (where transferability is hard, as noted in Ucar et al. [7]), overlapping columns seems to be what is needed to achieve transferability [24]. Getting overlapping columns seems to be nearly impossible between the three benchmark LTR public datasets, two of which (Yahoo, Istella) [25,26] do not specify what the columns even mean (due to the need to protect business trade secrets). The most transparent, MSLRWEB30K [27], also has many columns whose meanings may not make sense outside of the context of the dataset (for example, features that are the outputs of Microsoft internal models).

---

> > > > > > ### Author Response · Authors · 2023-11-22
> > > > > > **Follow up (references)**
> > > > > >
> > > > > > [1] Self-Supervised Contrastive Pre-Training For Time Series via Time-Frequency Consistency. Xiang Zhang, Ziyuan Zhao, Theodoros Tsiligkaridis, Marinka Zitnik. NeurIPS 2022.
> > > > > > [2] A Simple Framework for Contrastive Learning of Visual Representations. Ting Chen, Simon Kornblith, Mohammad Norouzi, Geoffrey Hinton. ICML 2020.
> > > > > > [3] Exploring Simple Siamese Representation Learning. Xinlei Chen, Kaiming He. CVPR 2021.
> > > > > > [4] CLOCS: Contrastive Learning of Cardiac Signals Across Space, Time, and Patients. Dani Kiyasseh, Tingting Zhu, David A. Clifton. ICML 2021.
> > > > > > [5] TS2Vec: Towards Universal Representation of Time Series. Zhihan Yue, Yujing wang, Juanyong Duan, Tianmeng Yang, Congrui Huang, Yunhai Tong, Bixiong Xu. AAAI 2022.
> > > > > > [6] Time-Series Representation Learning via Temporal and Contextual Contrasting. Emadeldeen Eldele, Mohamed Ragab, Zhenghua Chen, Min Wu, Chee Keong Kwoh, Xiaoli Li, Cuntai Guan. IJCAI 2021.
> > > > > > [7] SubTab: Subsetting Features of Tabular Data for Self-Supervised Representation Learning. Talip Ucar, Ehsan Hajiramezanali, Lindsay Edwards. NeurIPS 2021.
> > > > > > [8] Unsupervised Representation Learning by Predicting Image Rotations. Spyros Gidaris, Praveer Singh, Nikos Komodakis. ICLR 2018.
> > > > > > [9] Multi-Task Self-Supervised Visual Learning. Carl Doersch, Andrew Zisserman. ICCV 2017.
> > > > > > [10] BERT: Pre-training of Deep Bidirectional Transformers for Language Understanding. Jacob Devlin, Ming-Wei Chang, Kenton Lee, Kristina Toutanova.
> > > > > > [11] RoBERTa: A Robustly Optimized BERT Pretraining Approach. Yinhan Liu, Myle Ott, Naman Goyal, Jingfei Du, Mandar Joshi, Danqi Chen, Omer Levy, Mike Lewis, Luke Zettlemoyer, Veselin Stoyanov.
> > > > > > [12] Graph Contrastive Learning with Augmentations. Yuning You, Tianlong Chen, Yongduo Sui, Ting Chen, Zhangyang Wang, Yang Shen. NeurIPS 2020.
> > > > > > [13] Inductive Representation Learning on Large Graphs. Will Hamilton, Zhitao Ying, Jure Leskovec. NeurIPS 2017.
> > > > > > [14] Using Self-Supervised Learning Can Improve Model Robustness and Uncertainty. Dan Hendrycks, Mantas Mazeika, Saurav Kadavath, Dawn Song.
> > > > > > [15] To tune or not to tune? adapting pretrained representations to diverse tasks. Matthew E Peters, Sebastian Ruder, and Noah A Smith.
> > > > > > [16] VIME: Extending the Success of Self- and Semi-supervised Learning to Tabular Domain. Jinsung Yoon, Yao Zhang, James Jordon, Mihaela van der Schaar. NeurIPS 2020.
> > > > > > [17] Towards Domain-Agnostic Contrastive Learning. Vikas Verma, Minh-Thang Luong, Kenji Kawaguchi, Hieu Pham, Quoc V. Le. ICML 2021.
> > > > > > [18] Improvements that don't add up: ad-hoc retrieval results since 1998. Timothy G. Armstrong, Alistair Moffat, William Webber, Justin Zobel. CIKM 2009.
> > > > > > [19] Bootstrap Your Own Latent A New Approach to Self-Supervised Learning. Grill et al. NeurIPS 2020.
> > > > > > [20] Contrastive learning with hard negative samples. Joshua Robinson, Ching-Yao Chuang, Suvrit Sra, Stefanie Jegelka. ICLR 2021.
> > > > > > [21] Debiased Contrastive Learning. Ching-Yao Chuang, Joshua Robinson, Lin Yen-Chen, Antonio Torralba, Stefanie Jegelka. NeurIPS 2020.
> > > > > > [22] Robust Contrastive Learning Using Negative Samples with Diminished Semantics. Songwei Ge, Shlok Mishra, Chun-Liang Li, Haohan Wang, David Jacobs. NeurIPS 2021.
> > > > > > [23] SCARF: Self-Supervised Contrastive Learning using Random Feature Corruption. Dara Bahri, Heinrich Jiang, Yi Tay, Donald Metzler. ICLR 2022.
> > > > > > [24] Transfer Learning with Deep Tabular Models. Roman Levin, Valeriia Cherepanova, Avi Schwarzschild, Arpit Bansal, C. Bayan Bruss, Tom Goldstein, Andrew Gordon Wilson, Micah Goldblum. ICLR 2023.
> > > > > > [25] Yahoo! Learning to Rank Challenge Overview. Olivier Chapelle, Yi Chang.
> > > > > > [26] Fast Ranking with Additive Ensembles of Oblivious and Non-Oblivious Regression Trees. Domenico Dato et al..
> > > > > > [27] Introducing LETOR 4.0 Datasets. Tao Qin, Tie-Yan Liu.
> > > > > > [28] Critically Examining the "Neural Hype": Weak Baselines and the Additivity of Effectiveness Gains from Neural Ranking Models. Wei Yang, Kuang Lu, Peilin Yang, Jimmy Lin. SIGIR 2019.
> > > > > > [29] Are Neural Rankers still Outperformed by Gradient Boosted Decision Trees? Zhen Qin Le Yan Honglei Zhuang Yi Tay Rama Kumar Pasumarthi Xuanhui Wang Mike Bendersky Marc Najork. ICLR 2021.
> > > > > > [30] The Neural Hype and Comparisons Against Weak Baselines. Jimmy Lin. SIGIR 2019 Forums.

---

> > > > > > > ### Author Response · Authors · 2023-11-22
> > > > > > > **Gentle request for final feedback**
> > > > > > >
> > > > > > > Thank you for your time and consistent effort throughout this discussion! Your thoughtful comments have been very helpful in improving our paper. Today is the end of the discussion period, and we are eagerly waiting to hear any additional feedback you may still have about the paper.
> > > > > > >
> > > > > > > We hope that we have addressed your remaining concerns on (1) the novelty of the paper, and (2) transferability of results. Please let us know if there’s anything more we can do to improve our paper and/or increase our score.

---

### Meta-Review · Area_Chair_mnWT · 2023-12-06

**Metareview:**

The authors study tabular learning to rank (LTR) tasks. They focus specifically on a pretraining paradigm where they obtain good representations that are then used for training when labels are scarce. They establish two items: (i) there are cases where pretraining enables deep models to outperform tree-based methods and (ii) they propose a pretrained method that is cheaper than naively applying SimCLR, which is important.

The strengths are the overall findings. The weaknesses are various things identified by the reviewers: (i) the paper generally avoids (though has improved somewhat since the discussion phase) comparing a variety of baselines in favor of just comparing with GBDTs. (ii) how valuable is this particular finding by itself.

Overall I felt this work was borderline. I appreciate the contributions. To me, the weaknesses of the paper are the following: the authors identify a particular regime as the one of interest: few labels/lots of unlabeled data. They show a particular technique can outperform tree-based methods here, but they do not study a full range of possible techniques. For example, this is the area where semi-supervised techniques shine. One reviewer pointed this out, and the authors did one particular method (pseudolabeling)---but there has been almost 15 years of work on this area (i.e. Duh's dissertation "Learning to Rank with Partially-Labeled Data" is exactly on this topic). I'm not arguing that these methods are certain to outperform the authors' proposed variant of SimCLR, but it would be great to know.

The authors' argument for acceptance rests on two claims. First, they mention that their work is likely to have an outsized impact on actual applications. I believe this is possible, but it is very difficult to evaluate on our side. Second, they mention that the main finding is surprising and important. It is important, but it is also clear that if the label scarcity is severe enough, pretraining is virtually certain to outperform (i.e., the case of a handful of labels only).

One factor I did not take as an negative: technical novelty, which some reviewers brought up. I agree with the authors here that a very simple but efficient technique is valuable and novel.

Overall, I think this paper is clearly headed to an eventual acceptance (and perhaps the big impact the authors mention). However, I think a version of the paper that has a variety of baselines, that explores a variety of pretraining methods (also mentioned by the reviewers) and provides signal about what the right approach is, what the tradeoffs are between efficiency and seeking "better" hard negatives, etc would be the "right" version of this paper. As things stand it feels a bit incomplete.

While I'm sure this is disappointing, I do want to commend the authors for their expansive discussion with the reviewers.

**Justification For Why Not Higher Score:**

I describe some of this in the metareview. I would argue that the authors have done part of the work in discovering a regime where deep models (pretrained or perhaps otherwise) can outperform tree-based methods. It would be extremely valuable to produce an empirical study of this regime vs a variety of methods, baselines, etc. The paper feels slightly incomplete as-is.

**Justification For Why Not Lower Score:**

N/A

---

### Decision · Program_Chairs · 2024-01-16

Reject